# EvoImp: Multiple Imputation of Multi-label Classification data with a genetic algorithm

**Antonio Fernando Lavareda Jacob Junior**[1,2], **Fabricio Almeida do Carmo**[2], **Adamo Lima de Santana**[3], **Ewaldo Eder Carvalho Santana**[1,2], **Fabio Manoel Franca Lobato**[2,4]*

**1** Graduate Program in Electrical Engineering (PPGEE), Federal University of Maranhão (UFMA), São Luís, Maranhão, Brazil, **2** Graduate Program in Computer Engineering and Systems (PECS), State University of Maranhão (UEMA), São Luís, Maranhão, Brazil, **3** Corporate ReD Headquarters Fuji Electric Co., Tokyo, Japan, **4** Institute of Engineering and Geosciences, Federal University of Western Pará (UFOPA), Santarém, Pará, Brazil

* fabio.lobato@ufopa.edu.br

**Data Availability Statement:** All relevant files are available from the Zenodo database (Link: https://doi.org/10.5281/zenodo.7748933).

## Abstract

Missing data is a prevalent problem that requires attention, as most data analysis techniques are unable to handle it. This is particularly critical in Multi-Label Classification (MLC), where only a few studies have investigated missing data in this application domain. MLC differs from Single-Label Classification (SLC) by allowing an instance to be associated with multiple classes. Movie classification is a didactic example since it can be "drama" and "bibliography" simultaneously. One of the most usual missing data treatment methods is data imputation, which seeks plausible values to fill in the missing ones. In this scenario, we propose a novel imputation method based on a multi-objective genetic algorithm for optimizing multiple data imputations called Multiple Imputation of Multi-label Classification data with a genetic algorithm, or simply EvoImp. We applied the proposed method in multi-label learning and evaluated its performance using six synthetic databases, considering various missing values distribution scenarios. The method was compared with other state-of-the-art imputation strategies, such as K-Means Imputation (KMI) and weighted K-Nearest Neighbors Imputation (WKNNI). The results proved that the proposed method outperformed the baseline in all the scenarios by achieving the best evaluation measures considering the Exact Match, Accuracy, and Hamming Loss. The superior results were constant in different dataset domains and sizes, demonstrating the EvoImp robustness. Thus, EvoImp represents a feasible solution to missing data treatment for multi-label learning.

## Introduction

Missing data is ubiquitous in data analysis [1]. Their causes are the most diverse and related to the application domain. These include drawbacks in data acquisition, measurement errors, sensor network problems, data migration failures, and unwillingness to respond to survey questions [2, 3]. Since data analysis algorithms/methods are not designed to deal with Missing Values (MVs), it is essential to treat them before aiming to guarantee the results' validity, impairing the research conclusions [1, 4, 5]. MVs are problematic because of the risk of bias, which depends on the type of missing data, the extent of the missingness, and how to deal with

**Funding:** FMFL was financed in part by the National Council for Scientific and Technological Development (CNPq, Brazil) under Grant 147336/2020-1. The funders had no role in study design, data collection and analysis, decision to publish, or preparation of the manuscript.

**Competing interests:** The authors have declared that no competing interests exist.

MVs in the analyses [1]. Thus, it is critical to deal with the missing data timely for intelligent decision-making [6].

Several techniques have emerged to address this problem [4, 7, 8]. LIN [4] comments that if the MVs rate is less than 10% or 15%, they can be removed without causing any significant loss to the mining process. However, this does not mean that the datasets in any problem domain must follow this rule; in other words, small amounts of missing data may contain essential information that must be managed [9]. In addressing this issue, the literature suggests using missing data imputation methods, which involve replacing missing data with actual (plausible) values. While this approach allows for more data retention compared to deletion, it requires time to generate reasonable replacement values [10, 11].

A naïve method for tackling the missing values issue is by Single Imputation (SI). This method involves filling in missing values with a single estimated value, often based on mean, median, or regression models [4]. While this approach simplifies the dataset and makes it easier to analyze, it can introduce bias and underestimate uncertainty in the results [12, 13]. To overcome this limitation, Rubin [14] introduced a gold-standard imputation strategy within the scientific community—Multiple Imputations (MI) for handling missing data. In contrast with SI approaches, this method seeks to find a single solution in which $m$ complete solutions are created in the operational database such that $m > 1$. These solutions were analyzed separately and combined to obtain the best solution [15, 16]. To reduce the missing values prediction error, using metaheuristics could optimize the value that would be imputed [15]. Notably, bioinspired strategies such as Genetic Algorithms (GAs) are prominent in optimizing solutions [17].

The GAs were proposed by Holland [18]. It is an optimization heuristic based on "the survival of the fittest", inspired in Charles Darwin´s evolutionary theory. Regarding the GAs usage for Multiple Imputations, it is crucial to acknowledge the work of Garcia [19] and the MultImp algorithm [15]. The MultImp algorithm serves as the cornerstone for this research. This algorithm employed genetic algorithms for multiple imputations and was also applied for Multi-Label Classification (MLC) scenarios. The authors contend that data mining tasks, particularly those related to data classification, are notably sensitive to addressing MV. Furthermore, classification tasks are widely used to assess the accuracy (ACC) of imputation methods [5, 11, 20]. Consequently, the higher the classification accuracy, the more successful the imputation method. However, only a few studies have employed MLC. In contrast to Single-Label Classification (SLC), or simply data classification, which associates an example with a single label, MLC allows an instance to be associated with multiple labels, thereby increasing the complexity of classification tasks [21, 22]. Further details on this topic will be highlighted in the Background section.

Considering the importance of handling missing values in data analysis and the available solutions in the existing literature, this work presents an efficient algorithmic approach for multiple imputations applied to multi-label classification tasks. This method is named EvoImp, a combination of "*evolutionary*" and "*imputation*". Furthermore, the name is inspired by MultImp [15], which serves as the foundation for our algorithm and has shown promise in its preliminary stages for multiple imputations with missing data. EvoImp enhances the parameterization of MultImp to maximize its imputation capabilities and explores new configurations for computational experiments.

We conducted a rigorous benchmarking process to validate the proposed method's performance using diverse multi-label datasets. We compared EvoImp with well-established imputation methods documented in the literature. These datasets were systematically subjected to six missing value rates to simulate the Missing Completely At Random (MCAR) mechanism. The outcomes of these experiments were meticulously evaluated using five distinct classifiers. This

comprehensive evaluation provides insights into the strengths and potential limitations of our EvoImp when applied to real-world multi-label classification scenarios. By addressing the challenges associated with missing data in this context, our work aims to advance multi-label classification and the broader field of data analysis.

Accordingly, the remainder of this paper is organized as follows. The section "Background" presents a preliminary background. The section "EvoImp—Proposed Method" included the proposed method in this section. The section "Computational Experiments" details the experimental setup. The performance of the method and comparison with data imputation techniques are demonstrated in sections "Results and Analysis" and "Discussion". Finally, section "Conclusion and Suggestions for Future Work" summarizes the paper and points out potential directions for future exploration.

## Background

### Multi-label Classification and classical approaches to handling MVs

In single-label classification problems, a set of class labels is predetermined, and each object must be associated with one and only one label [23]. Formally, let $X$ denote the input/feature space, and $y$ denote the class value, where $y \in L$, which is the output space (a set of disjoint class labels). In this case, each sample is strictly associated with a single class label [24, 25]

However, there are increasingly more contexts in which data may belong to more than one class label. This classification condition is referred to as Multi-Label classification. Initially, MLC primarily focused on tasks such as text categorization, protein function classification, music categorization, semantic scene classification, and medical diagnosis [23, 24, 26]. Recently, new applications have emerged in Computer Vision, Natural Language Processing, and Data Mining, including Video Annotation, Legal Text Mining, and User Profiling [27]. According to [25, 28], similar to SLC, MLC is represented by $X$ and $y$, where each sample $x \in X$ is assigned a subset of the output space (a set of non-disjoint class labels). Table 1 illustrates a toy example depicting the difference between SLC and MLC, adapted from [29]. Considering that the data in Table 1 comprises 5 instances ($x_1, x_2, x_3, x_4, x_5$) and 3 labels ($y_1, y_2, y_3$).

Table 1a illustrates the SLC scenario, where five data instances ($x_1$ to $x_5$) are each strictly associated with a single label ($y_1$ to $y_3$). For instance, $x_1$ is associated with $y_1$, $x_2$ is associated with $y_2$, and so on. On the other hand, MLC allows data instances to be associated with

**Table 1. Comparison of SLC and MLC using a toy example with 5 instances and 3 labels.**

| Data | Label |
|---|---|
| $x_1$ | $y_1$ |
| $x_2$ | $y_2$ |
| $x_3$ | $y_3$ |
| $x_4$ | $y_1$ |
| $x_5$ | $y_3$ |
| (a) Single-label | |

| Data | Labels |
|---|---|
| $x_1$ | $y_1, y_2$ |
| $x_2$ | $y_2, y_3$ |
| $x_3$ | $y_1, y_3$ |
| $x_4$ | $y_2$ |
| $x_5$ | $y_3$ |
| (b) Multi-label | |

multiple labels simultaneously. Table 1b demonstrates the MLC scenario, where the same five data instances ($x_1$ to $x_5$) can have multiple labels assigned to them. For example, $x_1$ is associated with both $y_1$ and $y_2$, $x_2$ is associated with both $y_2$ and $y_3$, and so forth. This distinction highlights how SLC restricts each data instance to a single label, while MLC permits instances to belong to multiple labels simultaneously, making it more suitable for scenarios where objects or data points can be associated with different classes.

Although the difference is subtle in theory, MLC tends to be more challenging in practice. Gonçalves et al. [23] and Sá et al. [25] enumerated the following reasons for this:

- The possible classes of a given instance (output space) in MLC grow exponentially from the increasing number of labels. Therefore, when considering that a problem has $L$ distinct labels, the size of the output space in MLC is $2^L$ (combination of labels) while it is only $L$ in SLC;

- An MLC algorithm must consider whether there exists or not a correlation between labels. This kind of correlation is an essential step to ensure the effectiveness of several MLC processes [24, 30, 31];

- MLC systems performance evaluation uses different metrics than those traditionally used in SLC [32]. In SLC, the rating of a new instance can be either correct or wrong. On the other hand, in MLC, the result can be partially correct. It occurs when the classifier predicts some correct labels but includes some incorrect predictions or even omits a label that should be predicted. This problem requires cautious attention since some metrics follow contrasting aspects to define what is a good MLC prediction [25, 33];

- Unlike SLC problems, which traditionally involve the analysis of relational (structured) data, MLC applications typically address big data tasks, which involve semi-structured or unstructured data [24, 34].

All these challenges have amplified the complexity associated with handling MVs. Nevertheless, finding studies that relate MLC and MV is not straightforward, as demonstrated in [4, 8, 17].

In this context, we emphasize a limited number of studies that specifically address the issue of missing labels [35, 36], which means focusing on predicting an unknown label. Wang et al. [35] present a multi-label feature selection that considers feature interaction. For that, the authors use the definitions of multi-label neighborhood information entropy and multi-label neighborhood mutual information to mitigate the negative impact of missing labels. Cheng, Song & Qian [36] focus on addressing missing labels by leveraging label correlations and implementing a two-level kernel extreme learning machine autoencoder. The authors verified the proposed method on both missing and complete label datasets. Since these studies primarily focus on missing labels rather than missing values (predictive features), to the best of our knowledge, there is no work addressing missing values in the predictive feature space in an ML scenario. Thus, this constitutes one of the contributions of the present study.

## Bio-inspired computation for the handling of MVs

Tran, Zang, and Andreae [37] proposed a data imputation method by adopting an approach based on genetic programming called GPMI. An MI strategy was applied in this method, and an estimation of missing values was performed using regression techniques. The GPMI was compared with seven imputation methods through an experiment carried out in eight datasets and applying seven different missing values ratios (5, 10, 20, 30, 40, and 50) with the aid of

MCAR as a missing data mechanism. The classifier's accuracy was the performance measure adopted. The results suggest that the planned method performed better than all methods. According to the authors, genetic programming was primarily responsible for these results because the algorithm initially used random samples to fill the gaps before being submitted to genetic processes. The results confirmed that strategies based on evolutionary algorithms are feasible alternatives for missing values treatment.

Shahzad, Rehman, and Ahmed, in their study, "Missing Data Imputation using Genetic Algorithm for Supervised Learning" [38], employed GA to search for plausible values for missing data imputation. An exciting strategy adopted in this study is using information gain to observe how solutions are found as the process grows. In an experiment with five datasets that originally contained missing values, the proposed method was compared with other imputation approaches: the average, lowest value, highest value, zero, and MI. They used the following performance measures: predictive accuracy, precision, recall, F-measure, and the area under the Receiver Operating Characteristic (ROC) curve, with the following classifiers: NB-tree, PART, JRIP, Naive Bayes, KNN, and J48. The authors noted that the GA-based method showed promising results and worked well in datasets with a high percentage of missing values.

In [39], an algorithm called MOGAImp was proposed for multiple imputation datasets based on genetic algorithms. One of the exciting strategies of this work is to apply a multi-objective approach, which until then had not been adopted in the literature for the performance analysis of imputation techniques. This approach involves simultaneously employing two or more evaluation measures. It can be explained by the fact that there are distinctions between various performance measures because, while one increases, the other declines. In the case of MOGAImp, two conflicting measures were used: the classifier accuracy and the predictive accuracy of the imputation method, calculated using Normalized Root-Mean-Square error (NRMSE) and the Pareto front.

Another critical factor in the study conducted by [39] concerns population initialization, which employs a pool of candidate solutions based on each attribute. The solution pool involves grouping all possible dataset values for the attribute that has a missing value (by lexicographically comparing two strings in cases of categorical variables). The method was experimentally compared with other well-known techniques in the literature, employing benchmarking through several databases with missing values. The results demonstrated that the method achieved competitive performance and, according to the authors, demonstrated potential for real-world applications. However, high computational power is required for handling the MVs individually with MOGAImp and through the solution pool. Additionally, this strategy is an excellent alternative to a mixture of genetic materials. Therefore, it has been adopted in EvoImp as a baseline for mutation operations.

In [15], the authors created a scheme based on genetic algorithms, which served as a baseline for developing and analyzing the method employed in this study. The strategy, nominated as MultImp, predicts multiple imputations of datasets in a multi-label classification model. In this study, the authors conducted experiments using four databases that were initially completed. Subsequently, 5% of the missing values were added through the MCAR mechanism. Binary relevance (BR) was employed as the multi-label classifier, with C4.5 as a parameter. In the test scenario, MultImp was compared with two other imputation methods (K-Nearest Neighbors Imputation—KNNI and Most Common—MC) and evaluated lexicographically using the following measures: Exact Match (EM), Accuracy, and Hamming Loss (HL). The preliminary results of this study proved to be promising, particularly in the case of EM, where the performance achieved by the method was better in all the datasets used and justified adopting the lexicographical approach.

For a comprehensive summary of the works discussed in this section, we have provided a detailed table in our supplementary material, available on the project's GitHub repository (https://github.com/jacobjr/EvoImp).

## EvoImp—Proposed method

Since EvoImp is based on a genetic algorithm, the following descriptions explain how EvoImp was mapped and configured within the GA structure: a) the codification of individuals, b) the formation of the initial population, c) the configuration of genetic operators, and d) the definition of the fitness function. Fig 1 presents a toy example of this structure, which will be detailed in the following subsections.

### Individual encoding and population initialization

The individual encoding of EvoImp took place in the following form: the variables in the datasets represent individual genes. Genes initially marked with "?" represent the missing values (Fig 1(a)). Each individual is represented by a completed ("accomplished") instance of the databases (Fig 1(b)). The phenotype consists of imputed values, while the genotype represents these values in binary form, as illustrated in Fig 1(c).

The initial population comprised five simple imputation methods for the generation of each individual (Fig 1(d)). All imputation methods are well-known and established in the literature [7]: K-means Clustering Imputation, KNNI, WKNNI, Concept Most Common (CMC), and MC. The parameters for the KNNI, WKNNI, and KMI methods followed the guidelines set by the authors. This kind of population initialization was adopted in EvoImp to reduce the search space and, hence, the computational costs.

The methods employed are as follows [7]:

- **KNNI:** Whenever there is a missing value, the K-nearest neighbors closest to the instance containing the MV are determined. The most common value among the K-nearest neighbors was used to impute nominal attributes. For numerical attributes, imputation is performed by calculating the average of the neighboring values;

- **WKNNI:** This technique involves determining the distances between K-nearest neighbors and a weighting distribution regarding the distances between each neighbor. After this, the KNNI process was repeated;

- **KMI:** This technique divides a database into clusters based on their features. Once this has been done, the K-nearest neighbors technique is applied when deciding which value should be imputed;

- **MC:** In this method, the most common value is adopted for imputation in nominal attributes and the average of all corresponding attributes in the case of numerical attributes;

- **CMC:** This method does the same thing as MC but only employs the referenced attribute class with MV.

In contrast to MOGAImp [39], which employs random initialization of the initial population, the proposed method optimizes simple imputations through evolutionary processes to perform multiple imputations. This approach reduces the search space and introduces a novel method. This reduction in search space is particularly beneficial in scenarios where computational cost is critical in objective function calculations, such as multi-label classification.

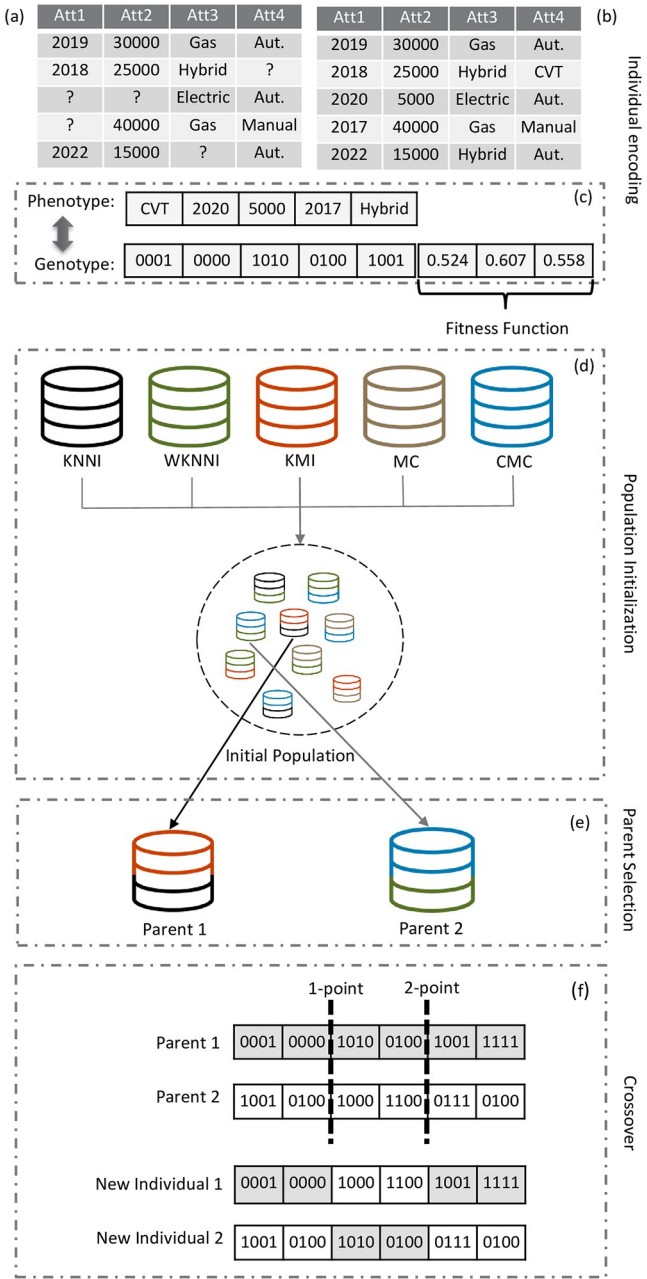

**Fig 1. EvoImp's GA structure example.** Toy example of a dataset with MV and how EvoImp's GA works with it. (a) Dataset with missing values; (b) A complete dataset with imputed data. (c) Phenotype: contains the values corresponding to the missing data space; Genotype: represents the genes in binary code and the values of the measurements used in the fitness function. (d) Illustration of how the initial population is initialized. (e) Random selection of parents for crossover. (f) Illustration of crossover being applied to the two selected individuals.

It is also noteworthy that the presented work has two innovative contents: 1) using simple imputation methods as *a priori* solution, reducing the search space; 2) treating missing values in the multi-label scenario. To our best knowledge, there is no similar study in the literature.

## Genetic operators

The individual selection involves a tournament in which two (or more) members of the previous population are selected, and the better one is chosen based on fitness value, as illustrated in Fig 1(e). This procedure was followed until a limited number of individuals from the current generation were obtained. The best individual is always selected through elitism [40].

In the literature, numerous methods for parameter tuning and control have been proposed and analyzed. [41] describes some of these methods and discusses various trends and challenges in the field. Specifically, [42] conducted experiments to find appropriate settings for these parameters when applying evolutionary algorithms to a multi-objective problem class. They concluded that determining the value of the scaling factor can be difficult and is highly dependent on the specific problem. Considering these findings, initial tests were conducted to define the parameters used in our study. In line with the work of [42], the initial percentage of Crossover was delimited to [0.8, 1.0], following the standard proposal for non-separable problems like the one tackled in our research. EvoImp employs a crossover for 80% of the individuals using an n-point crossover operator [43], as shown in Fig 1(f). It is also consonant with the work of [44].

The mutation process is performed on 20% of the individuals chosen randomly, except for the best one. For each individual to be mutated, the imputed value is exchanged for a candidate value. The mutation is applied only to genes that contain missing values. To accomplish this, each attribute in the dataset has a set of solutions, as shown in Table 2. This set is formed by considering all possible response options for that attribute in the evaluated dataset.

Table 2a displays a toy dataset containing five records and four attributes: "Year", "Gender", "Age", and "Have Credit". Some values in the dataset are missing and are represented by "?". Table 2b lists the possible values for each attribute. For example, the "Year" attribute can have values 1998, 2005, or 2010; and the "Gender" attribute can have values M or F. The same reasoning is applied to the other attributes.

Lobato et al. [39] adopted this technique to initiate the first MOGAImp population. The mutation operator was not implemented in MultImp. The lack of it caused a premature convergence, limiting the method's robustness. That operator is one of the main differences between MultImp and EvoImp. In other words, the proposed method implements a strategy to avoid local minimum.

The algorithm's search and optimization process occurs over predetermined generations. The population goes into a growth phase, starting with the number of MI methods adopted in

**Table 2. Candidate solutions for each attribute used for the mutation process.**

| Year | Gender | Age | Have Credit |
|------|--------|-----|-------------|
| 2010 | M | 25 | ? |
| ? | F | ? | Yes |
| 2005 | ? | 32 | ? |
| 1998 | M | ? | Yes |
| ? | ? | 30 | No |

(a) Toy Example of a dataset with MV.

| Attribute | Values |
|-----------|--------|
| Year | 1998, 2005, 2010 |
| Gender | M, F |
| Age | 25, 30, 32 |
| Have Credit | Yes, No |

(b) Set of possible values

the population initialization and increasing by its cross-over. This strategy aims to provide population diversity. In the second phase, the population is gradually reduced, achieving the same initial population size, allowing the analysis to choose the best solution qualitatively.

## Fitness function

As mentioned earlier, the method was evaluated on an MLC scenario. For this, EvoImp performs a classification process on each individual. The goal is to analyze the performance of the classifier and, consequently, the data imputation efficiency.

Three performance measures were adopted to evaluate the classifier, as with MultImp: Exact Match, Accuracy, and Hamming Loss. The notation used by [15, 45] were adopted to describe these measures: (i) $n$: number of instances in the test set; (ii) $q$: number of labels; (iii) $Y_i$: set of original labels, for instance, $i$; and (iv) $Z_i$: set of predictive labels, for instance, $i$.

- **Exact Match** calculates, using a binary system, whether all the instance labels are predicted correctly. This measure, as expressed in Eq 1, is assumed to be trivial because it ignores partial predictions:

$$EM = \frac{1}{n}\sum_{i=1}^{n} I(Y_i = Z_i)$$ (1)

- **Accuracy** is also a measure that counts the correctly predicted labels of an instance. In this case, partial predictions are taken into account. Eq 2 expresses the mathematical model of this measure:

$$ACC = \frac{1}{n}\sum_{i=1}^{n} \frac{Y_i \cap Z_i}{Y_i \cup Z_i}$$ (2)

- **Hamming loss** is a measure that, in contrast to *accuracy*, evaluates the classifier's performance by finding the average of incorrect predictions. Eq 3 describes this measure:

$$HL = \frac{1}{n}\sum_{i=1}^{n} \frac{Y_i \Delta Z_i}{q}$$ (3)

These measures were used in lexicographical order; in other words, this approach prioritizes all the problem's objectives and then tries to satisfy them, keeping a list of priorities [46]. Thus, the fitness ($f$) for the problem solution can be expressed as Eq 4:

$$f = [f_0, f_1, ..., f_{n-1}] \in \mathbb{R}^n$$ (4)

where $n$ is the number of objectives defined; $f_n$ is an optimization goal. Given two fitness evaluations $f_1$ and $f_2$ and a precision threshold $t$, the lexicographic relation between them (noted as $\prec_l$ and $\preceq_l$) can be defined [47]:

$$f_1 \prec_l f_2 \Leftrightarrow \exists k \in [0, n_0) \cap \mathbb{N} : f_1^k < f_2^k \wedge |f_1^k - f_2^k| \geqslant t$$
$$\wedge |f_1^i - f_2^i| < t \; \forall i < k$$ (5)

$$f_1 =_l f_2 \Leftrightarrow |f_1^i - f_2^i| < t \ \forall i \in [0, n_o) \cap \mathbb{N} \tag{6}$$

$$f_1 \preceq_l f_2 \Leftrightarrow f_1 \prec_l f_2 \ \lor f_1 =_l f_2 \tag{7}$$

As can be observed, the Eq 5 shows $f_1 \prec_l f_2$, which means that $f_1$ is lexicographically less than $f_2$. This relationship is established when there exists an index $k$ in in the range $[0, n_0) \cap \mathbb{N}$, such that $f_1^k < f_2^k$, indicating that the $k$-th component of $f_1$ is less than the $k$-th component of $f_2$. Additionally, the difference between $f_1^k$ and $f_2^k$ is greater than or equal to $t$. This ensures that the $k$-th components differ significantly by at least $t$. Finally, the absolute differences between corresponding components $f_1^i$ and $f_2^i$ should be less than $t$ for all $i$ less then $k$. In essence, this relation means that $f_1$ is superior to $f_2$ in terms of some objectives. The Eq 6 determines equality in lexicographical order ($f_1 =_l f_2$). This occurs when the absolute differences between corresponding components $f_1^i$ and $f_2^i$ are all less than $t$ for all $i$ in the range $[0, n_0) \cap \mathbb{N}$. In other words, $f_1$ and $f_2$ are considered equal regarding their performance across objectives. Finally, the Eq 7 presents $f_1 \preceq_l f_2$, which means that $f_1$ is either less than or equal to $f_2$ in lexicographical order. It combines the $\prec_l$ and $\preceq_l$ relations, indicating that $f_1$ is either better than or equal to $f_2$ in terms of the defined objectives.

These equations are used to rank and compare solutions or fitness evaluations in optimization problems, considering the objectives, prioritization, and performance. The lexicographical order approach allows for precise, multi-objective optimization when there are multiple criteria or objectives to be considered. Once the threshold $t$ has been introduced, this formulation differs from the pure mathematical lexicographic relation. It permits the decision maker to choose the precision to compare two fitness functions. This relation allows the ranking of solutions of EvoImp as follows:

1. The EM behavior is evaluated;

2. If two or more individuals match their respective scores, the ACC evaluation is checked;

3. If the tie remains, the HL evaluation is used.

This approach allows different performance measures to be added to a single evaluation [45]. It is similar to the classical lexicographical approach, but once evolutionary algorithms are adopted, local optima can be avoided [47].

## The EvoImp algorithm

As shown in Algorithm 1, EvoImp begins the execution by creating and evaluating individuals for the initial population. The datasets are initially imputed using simple imputation methods: KNNI, CMC, MC, KMI, and WKNNI (lines 1–5). Afterward, the population is evaluated and ranked based on each individual's performance (line 6). The algorithm applies the genetic operators if the stopping criterion is not attained (e.g., the number of generations).

Algorithm 1: EvoImp

```
Input: datasets with MV and parameters (see Table 4)
Output: complete datasets
1 foreach Simple Imputation Method do
2    Generate a new Individual: individual;
3    Evaluate the individual;
4    Add individual to the Current Population: currentPop ←
     individual;
5 end
6 Order currentPop using Lexicographical order;
7 while Stop criterion not reached do
```

```
8   Add to the Current population the Best Individual: currentPop ←
    bestIndividual;
9   while currentPop < Number individuals of the new generation do
10     Select Parents;
11     Appy Crossover;
12     Evaluate the new Individual;
13     Add the Individual to Current population: currentPop ←
       individual;
14  end
15  while Number of mutated individuals < 20% of Individuals of the
           new generation do
16     Randomly choose an individual from the Current population;
17     Apply Mutation;
18     Evaluate the Individual;
19     Add the Individual to Current population: currentPop ←
       individual;
20  end
21  Order currentPop using Lexicographical order;
22 end
23 return bestIndividual;
```

The elitist individual is always passed on to the next generation (line 8). The selection is performed using the tournament selection operator (line 10). Two individuals are randomly drawn in this process. These two parents exchange genetic material using a crossover operator. These steps are repeated until the population is complete. Afterward, the mutation follows the established rate (lines 15–20). The new population is arranged, and the iterative process continues until the stopping criterion is reached. The return of the algorithm is the individual that achieves the best performance (line 23).

In summary, EvoImp adopts the configuration for the parameterization of MultImp [15], except for the mutation operator, as pointed out earlier. Besides, we also corrected bugs and optimized the code, bearing in mind maintainability and reuse. Moreover, we implemented the lexicographic strategy and expanded the computational tests, expanding the technical-scientific contribution of the present work.

## Computational experiments

### Datasets

The experiments were designed using six multi-label datasets from the UCI Machine Learning repository (https://archive.ics.uci.edu/). The quantity datasets agree with the literature review conducted by [17], which mapped 48 papers related to experiments in the context of data imputation. Chiu's work [17] shows that most papers (77%) use up to six datasets in experiments. Another interesting finding of Chiu et al. [17] is that the UCI Machine Learning Repository is the most used. Regarding the characteristics of the datasets, most use small-scale datasets, which contain fewer than 15 attributes and 800 instances. Table 3 shows the datasets used and their features.

Regarding multi-label datasets, the works of [35, 48] must be mentioned. These studies, as well as EvoImp, used datasets obtained at the UCI repository and formatted using the Mulan library (http://mulan.sourceforge.net/). The datasets used in these papers have similar characteristics (cardinality, density, and the number of instances) to those chosen in this paper. This observation highlights the experimental setup consonance with the state of the art and the EvoImp potential applicability in real-world problems.

**Table 3. Datasets used in experiments.**

| Dataset | Cod. | Domain | Inst. | Nominal Atr. | Numerical Atr. | Total Atr. | Labels | Cardinality | Density |
|---------|------|--------|-------|--------------|----------------|------------|--------|-------------|---------|
| Birds | b | Audio | 645 | 2 | 258 | 260 | 19 | 1.014 | 0.053 |
| Cal500 | c | Music | 502 | 0 | 68 | 68 | 174 | 26.044 | 0.150 |
| Emotions | e | Music | 593 | 0 | 72 | 72 | 6 | 1.869 | 0.311 |
| Flags | f | Image | 194 | 9 | 10 | 19 | 7 | 3.392 | 0.485 |
| Scene | s | Image | 2407 | 0 | 294 | 294 | 6 | 1.074 | 0.179 |
| Yeast | y | Biology | 2417 | 0 | 103 | 103 | 14 | 4.237 | 0.303 |

## Experimental setup

In the experiments, the missing values were artificially added to each dataset with the following rates: 5%, 10%, 15%, 20%, 25%, and 30%. This "amputation" process was carried out using the MCAR mechanism, as described in Santos (2019) [49]. The complete experimental configuration consisted of 36 datasetss with missing data, and these datasetss underwent a comparative evaluation. This evaluation involved five simple imputation methods: KNNI, CMC, MC, KMI, and WKNNI.

The following classification methods were used for the multi-label learning tasks: Binary Relevance (BR), Hierarchy of Multi-label classifiER (HOMER), Multi-Label K-Nearest Neighbors (ML-KNN), Classifier Chains (CC), and Ensembles of Classifier Chains (ECC) [21, 50]. K-fold cross-validation was used for the classification model's evaluation (learning and testing). Table 4 summarizes the overall parameters which were used in the experiments.

Regarding the simple imputation methods, the parameters recommended by [7] were used. The mutation rate (MR) chosen is higher than the typical usage rates because the starting point is not random. Therefore, considering that the initial population is obtained by other methods, parameterization experiments demonstrated that a higher MR yields better results, providing fast convergence. The entire experimental setup and the obtained results are available as supplementary material on the project's GitHub (https://github.com/jacobjr/EvoImp).

## Implementation

The GA was programmed in the Java language, version 8.1, based on the works of [15, 44]. Other components used third-party implementations as follows:

**Table 4. Parameter settings used in the experiments.**

| Parameter | Value |
|-----------|-------|
| Initial population | five individuals (imputed datasets) |
| Generations | 7 |
| Crossover rate | 80% of individuals |
| Mutation rate | 20% |
| Selection type | Tournament (size = 2) |
| Imputation methods | KNNI, CMC, MC, KMI, and WKNNI |
| MV rates | 5, 10, 15, 20, 25, and 30% |
| Method of MV occurrence | MCAR |
| MLC algorithms | BR, HOMER, ML-KNN, CC, and ECC |
| K-fold cross-validation | k = 10 |

- For the multi-label classifiers, Mulan's library (https://mulan.sourceforge.net/) was used [51]. This library also contains some classifiers implemented in Weka (https://www.cs.waikato.ac.nz/ml/weka/index.html) [52].

- The simple imputation methods used for forming the first population of EvoImp and in the comparative analyses are implemented in KEEL-software (http://www.keel.es/) [53].

It is noteworthy that GA used in the EvoImp was fully implemented by the authors despite KEEL providing a framework for evolutionary computation. This design decision aimed to give us more control over the experiments. The computational complexity is another crucial aspect to consider in implementing this proposed method. It plays a vital role in determining the feasibility and efficiency of applying bio-inspired techniques to solve optimization problems. Addressing this concern and reducing computational complexity enhances the algorithm's applicability and scalability. As a result, it makes it more suitable for handling larger datasets and complex optimization landscapes, particularly in multi-label classification tasks. More detailed information about EvoImp's computational complexity can be found in the supplementary materials on the project's GitHub repository.

## Results and analysis

This section examines the results obtained from the computational experiments. The data displayed in the following tables show the differences in performance between the methods for each percentage of missing values analyzed (5%, 10%, 15%, 20%, 25%, and 30%). The best results are highlighted in bold for easy viewing. The metrics (Exact Match (↑), Accuracy (↑), and Hamming Loss (↓)) are presented with these symbols, where (↑) indicates that higher values reflect better performance, and (↓) indicates that lower values represent better performance.

### Binary relevance

In the learning performed with the BR classifier, the results showed that the EvoImp was numerically superior (Table 5). In the EM evaluation, EvoImp outperformed its competitors in 35 of the 36 datasets evaluated (97.22%). The proposed method demonstrated superior performance compared to others in 18 scenario datasetss (50%) regarding the Accuracy evaluation measure. Finally, considering the HL, EvoImp outperformed the baseline methods in 16 datasets (44.44%).

It is essential to highlight the priorities adopted in the EvoImp lexicographic order, prioritizing the evaluation with EM, as mentioned in the Subsection "Fitness Function", which explains the performance decrease for the ACC and HL metrics considering the binary relevance classifier.

### Hierarchy Of Multi-label Classifier (HOMER)

The results for the HOMER classifier are given in presented in Table 6. Analyzing the results, it is possible to observe that EvoImp is also superior to the others in 35 of the 36 datasets used in the experiments (97.22%) regarding the EM metric. These results corroborate the ones obtained from the Binary Relevance classifier.

Continuing analyzing Table 6 results, regarding the ACC evaluation measure, EvoImp outperformed the baseline methods in 23 datasets (63.88%). The HL results show that EvoImp had the slightest error in classification in 19 out of 36 datasets (52.78%). In summary, EvoImp outperformed the methods for all performance measures for HOMER classifier, in consonance with the results for BR classifier as well.

**Table 5. Experimental results for the binary relevance classifier.**

| %[1] | Db[2] | Exact Match (↑) | | | | | | Accuracy (↑) | | | | | | Hamming Loss (↓) | | | | | |
|---|---|---|---|---|---|---|---|---|---|---|---|---|---|---|---|---|---|---|---|
| | | KMI[3] | KNNI[3] | MC[3] | CMC[3] | WKNNI[3] | EvoImp | KMI | KNNI | MC | CMC | WKNNI | EvoImp | KMI | KNNI | MC | CMC | WKNNI | EvoImp |
| 5 | b | 46.67 | 48.54 | 51.32 | 51.22 | 51.31 | 52.57 | 58.01 | 61.82 | 62.33 | 60.91 | 62.32 | 62.94 | 05.52 | 05.21 | 05.05 | 05.06 | 05.13 | 05.01 |
| | c | 33.77 | 34.92 | 33.93 | 34.13 | 34.86 | 35.24 | 44.04 | 44.88 | 44.87 | 45.12 | 44.75 | 44.73 | 15.81 | 15.92 | 15.55 | 15.42 | 15.91 | 15.87 |
| | e | 50.78 | 50.82 | 48.83 | 51.21 | 50.71 | 51.44 | 54.04 | 56.74 | 53.42 | 56.77 | 56.54 | 57.24 | 24.92 | 25.02 | 25.94 | 24.78 | 25.32 | 24.51 |
| | f | 58.82 | 58.88 | 60.22 | 59.41 | 58.44 | 61.94 | 70.02 | 68.75 | 69.82 | 69.51 | 68.63 | 68.48 | 27.42 | 28.49 | 27.86 | 28.03 | 28.63 | 27.76 |
| | s | 53.59 | 56.92 | 51.04 | 52.81 | 56.93 | 58.12 | 51.23 | 55.85 | 48.84 | 50.71 | 55.27 | 56.68 | 14.14 | 13.34 | 14.67 | 13.93 | 13.31 | 13.02 |
| | y | 50.67 | 49.72 | 50.17 | 50.53 | 50.64 | 51.32 | 59.90 | 59.92 | 60.33 | 61.17 | 59.94 | 61.78 | 24.83 | 24.96 | 24.22 | 23.92 | 24.89 | 23.84 |
| 10 | b | 46.81 | 48.43 | 47.71 | 46.82 | 46.33 | 49.87 | 56.51 | 60.02 | 60.24 | 56.93 | 57.42 | 60.11 | 05.74 | 05.32 | 05.12 | 05.44 | 05.51 | 05.32 |
| | c | 35.54 | 36.14 | 34.71 | 34.63 | 36.46 | 36.63 | 44.45 | 45.62 | 45.36 | 45.22 | 45.78 | 45.85 | 15.52 | 15.53 | 14.92 | 15.04 | 15.52 | 15.51 |
| | e | 47.54 | 49.87 | 44.82 | 49.63 | 48.25 | 50.97 | 53.62 | 53.24 | 47.04 | 52.27 | 54.43 | 54.12 | 26.72 | 26.05 | 28.13 | 25.22 | 26.56 | 25.96 |
| | f | 57.52 | 60.22 | 57.43 | 58.41 | 60.34 | 61.64 | 70.43 | 73.12 | 70.99 | 68.73 | 73.34 | 73.72 | 26.93 | 26.13 | 26.22 | 28.31 | 25.96 | 25.22 |
| | s | 52.65 | 56.81 | 47.68 | 51.82 | 57.03 | 58.34 | 51.12 | 56.13 | 43.61 | 48.16 | 56.56 | 57.37 | 13.94 | 12.94 | 15.25 | 13.91 | 12.85 | 12.34 |
| | y | 49.56 | 50.64 | 48.81 | 48.42 | 50.22 | 51.93 | 59.23 | 60.31 | 59.66 | 59.13 | 60.37 | 61.17 | 24.85 | 24.59 | 24.21 | 24.42 | 24.67 | 24.43 |
| 15 | b | 44.42 | 45.87 | 47.32 | 43.86 | 44.62 | 47.36 | 56.31 | 57.38 | 59.03 | 54.58 | 54.52 | 59.14 | 05.63 | 05.26 | 04.98 | 05.62 | 05.31 | 04.92 |
| | c | 34.73 | 35.41 | 36.06 | 36.01 | 35.95 | 36.17 | 43.91 | 43.97 | 47.03 | 47.55 | 44.72 | 47.36 | 14.92 | 15.43 | 14.19 | 14.04 | 15.32 | 14.08 |
| | e | 48.41 | 49.43 | 44.09 | 47.61 | 48.13 | 50.32 | 52.73 | 55.32 | 46.63 | 53.51 | 55.74 | 56.22 | 26.31 | 25.59 | 27.05 | 25.13 | 24.92 | 25.46 |
| | f | 61.74 | 61.76 | 59.03 | 61.85 | 63.71 | 64.35 | 73.11 | 73.45 | 72.14 | 72.16 | 74.95 | 75.07 | 25.91 | 24.46 | 25.95 | 24.98 | 23.06 | 22.87 |
| | s | 50.72 | 58.16 | 46.35 | 49.64 | 58.19 | 58.95 | 48.94 | 57.95 | 41.51 | 47.15 | 57.14 | 57.65 | 14.24 | 12.56 | 15.41 | 13.54 | 12.51 | 12.21 |
| | y | 47.50 | 50.94 | 48.71 | 49.95 | 50.94 | 51.44 | 57.61 | 60.25 | 60.27 | 60.76 | 59.91 | 61.09 | 24.41 | 24.34 | 23.36 | 24.42 | 24.74 | 24.38 |
| 20 | b | 43.04 | 45.84 | 43.28 | 42.51 | 43.79 | 47.37 | 54.66 | 58.27 | 53.94 | 52.26 | 55.51 | 57.65 | 05.51 | 04.86 | 05.15 | 05.52 | 05.20 | 04.73 |
| | c | 35.82 | 35.41 | 35.55 | 35.47 | 35.45 | 36.45 | 44.02 | 43.84 | 47.36 | 46.38 | 43.43 | 44.31 | 14.44 | 15.17 | 13.51 | 13.64 | 15.27 | 14.94 |
| | e | 45.44 | 45.44 | 40.82 | 48.46 | 46.23 | 48.92 | 52.38 | 50.34 | 39.94 | 51.17 | 50.49 | 51.24 | 25.36 | 27.28 | 27.51 | 24.04 | 27.42 | 23.85 |
| | f | 57.93 | 61.05 | 59.61 | 62.51 | 60.25 | 63.37 | 70.02 | 71.76 | 71.74 | 71.73 | 72.47 | 72.12 | 27.14 | 24.46 | 26.04 | 24.97 | 23.92 | 24.56 |
| | s | 45.52 | 57.74 | 43.61 | 47.48 | 58.45 | 58.74 | 42.01 | 57.26 | 39.08 | 44.34 | 57.63 | 57.73 | 15.23 | 12.36 | 15.48 | 13.69 | 11.85 | 11.72 |
| | y | 49.25 | 50.92 | 50.16 | 49.23 | 51.57 | 51.78 | 58.53 | 61.06 | 61.67 | 61.75 | 60.85 | 60.82 | 24.52 | 24.07 | 22.05 | 22.36 | 23.92 | 23.29 |
| 25 | b | 43.45 | 43.44 | 44.15 | 42.14 | 43.94 | 44.75 | 55.47 | 56.25 | 56.56 | 52.57 | 57.78 | 56.91 | 05.15 | 05.01 | 04.84 | 05.01 | 04.83 | 04.73 |
| | c | 37.76 | 37.21 | 37.75 | 36.26 | 36.47 | 37.85 | 45.81 | 44.85 | 50.56 | 47.77 | 44.21 | 50.56 | 13.94 | 14.98 | 12.64 | 13.01 | 15.06 | 12.62 |
| | e | 43.23 | 45.14 | 38.22 | 46.57 | 45.06 | 47.72 | 44.35 | 50.13 | 37.21 | 49.32 | 49.41 | 50.12 | 25.94 | 26.82 | 26.31 | 23.48 | 27.09 | 25.24 |
| | f | 60.25 | 58.92 | 62.23 | 60.24 | 59.71 | 63.46 | 72.51 | 72.65 | 73.13 | 70.32 | 72.47 | 73.04 | 27.04 | 25.83 | 26.08 | 27.42 | 25.15 | 25.31 |
| | s | 47.14 | 59.36 | 39.94 | 45.16 | 58.87 | 60.16 | 43.34 | 58.68 | 36.53 | 42.32 | 58.74 | 59.14 | 14.61 | 11.73 | 15.48 | 13.61 | 12.09 | 11.72 |
| | y | 49.72 | 51.75 | 48.51 | 49.23 | 51.21 | 51.96 | 62.96 | 61.65 | 61.74 | 61.71 | 61.08 | 61.74 | 24.68 | 23.21 | 22.05 | 21.84 | 23.68 | 23.29 |
| 30 | b | 39.15 | 39.41 | 42.27 | 42.31 | 41.03 | 43.36 | 50.25 | 51.71 | 52.82 | 52.76 | 53.48 | 53.03 | 05.51 | 05.48 | 04.94 | 05.22 | 05.21 | 05.14 |
| | c | 37.13 | 35.92 | 37.82 | 37.65 | 35.43 | 38.01 | 45.21 | 43.24 | 49.92 | 49.01 | 43.26 | 47.57 | 13.62 | 14.73 | 12.02 | 12.21 | 14.62 | 12.97 |
| | e | 44.52 | 45.96 | 41.12 | 48.91 | 48.03 | 49.62 | 48.63 | 49.22 | 38.85 | 53.67 | 53.65 | 54.37 | 26.52 | 26.81 | 26.24 | 24.06 | 26.13 | 23.81 |
| | f | 62.04 | 62.42 | 59.71 | 64.74 | 63.11 | 65.54 | 74.32 | 74.23 | 74.74 | 75.13 | 74.58 | 75.46 | 24.68 | 24.42 | 24.84 | 23.72 | 24.73 | 23.26 |
| | s | 44.53 | 59.32 | 39.46 | 44.18 | 58.97 | 59.62 | 40.94 | 59.01 | 35.78 | 41.92 | 59.38 | 59.23 | 15.37 | 11.79 | 15.32 | 13.52 | 11.74 | 11.62 |
| | y | 50.13 | 51.82 | 49.71 | 49.04 | 51.57 | 52.71 | 59.96 | 62.23 | 62.72 | 63.02 | 62.04 | 62.84 | 23.52 | 22.83 | 20.92 | 21.04 | 22.94 | 22.52 |
| Avg rank | – | 5 | 2 | 6 | 4 | 3 | 1 | 6 | 3 | 5 | 4 | 2 | 1 | 6 | 5 | 3 | 2 | 4 | 1 |

[1]"%" refers to the percentage of missing data analyzed (5%, 10%, 15%, 20%, 25%, and 30%).

[2]"Db" refers to the datasets used in the experimental setup, and these letters' abbreviations can be found in Table 3.

[3]Acronyms are related to each data imputation method tested, listed in S1 Table. Abbreviations.

**Table 6. Experimental results for HOMER classifier.**

| % [1] | Db [2] | Exact Match (↑) KMI [3] | KNNI [3] | MC [3] | CMC [3] | WKNNI [3] | EvoImp | Accuracy (↑) KMI | KNNI | MC | CMC | WKNNI | EvoImp | Hamming Loss (↓) KMI | KNNI | MC | CMC | WKNNI | EvoImp |
|---|---|---|---|---|---|---|---|---|---|---|---|---|---|---|---|---|---|---|---|
| 5 | b | 43.83 | 48.63 | 49.72 | 48.13 | 49.21 | **52.47** | 54.05 | 59.51 | 58.92 | 56.42 | 60.03 | **60.72** | 06.62 | 05.96 | 05.84 | 06.12 | 06.25 | **05.58** |
| | c | 36.43 | 36.41 | 35.36 | 36.48 | 36.54 | **37.93** | 35.01 | 35.64 | 34.42 | 34.74 | 35.21 | **35.56** | 20.42 | **20.21** | 20.52 | 20.45 | 20.47 | 20.39 |
| | e | 47.12 | 51.04 | 46.72 | 51.12 | 49.37 | **53.05** | 51.92 | 54.96 | 50.70 | 55.73 | 53.62 | **56.31** | 26.64 | 25.62 | 26.43 | 25.28 | 26.41 | **24.85** |
| | f | 61.32 | 61.53 | 60.69 | 60.02 | 60.23 | **63.02** | 68.25 | 67.72 | 67.71 | 66.04 | 67.82 | **68.42** | 27.33 | 27.83 | 27.72 | 29.12 | 27.51 | **27.28** |
| | s | 51.92 | 54.47 | 50.43 | 51.72 | 55.04 | **55.32** | 48.78 | 52.66 | 47.95 | 48.83 | 53.12 | **53.35** | 14.90 | 14.26 | 15.22 | 14.62 | 14.04 | **14.01** |
| | y | 50.42 | 50.12 | 48.97 | 50.04 | 50.79 | **50.92** | 58.24 | 58.63 | 57.12 | 56.94 | **58.82** | 58.64 | 25.98 | 25.70 | 26.53 | 26.41 | **26.02** | 26.13 |
| 10 | b | 44.92 | 46.96 | 45.29 | 43.16 | 47.72 | **48.25** | 55.32 | 56.41 | 57.09 | 52.73 | 57.26 | **57.74** | 06.52 | 06.42 | 06.15 | 06.43 | 06.12 | **06.04** |
| | c | 35.72 | 36.84 | 35.93 | 36.92 | 36.92 | **37.95** | 34.33 | 36.38 | 34.62 | 34.95 | 36.03 | **36.75** | 20.03 | 19.62 | 19.66 | 19.62 | 19.71 | **19.52** |
| | e | 48.9 | 48.83 | 43.74 | 49.54 | 49.62 | **50.64** | 53.46 | 53.32 | 47.43 | 52.21 | 53.74 | **53.95** | 26.21 | 26.72 | 27.06 | 26.83 | 26.68 | 26.32 |
| | f | 60.74 | 60.32 | 58.93 | 60.74 | 60.48 | **61.23** | 68.94 | 71.52 | 67.73 | 67.71 | **69.92** | 67.87 | 27.42 | **25.93** | 27.64 | 27.92 | 27.01 | 27.74 |
| | s | 52.12 | 55.23 | 47.05 | 47.79 | 55.38 | **55.72** | 49.44 | 53.23 | 43.18 | 44.08 | **53.56** | 53.53 | 14.35 | 13.78 | 16.02 | 15.64 | 13.82 | **13.71** |
| | y | 50.54 | 50.52 | 50.31 | 48.95 | 50.03 | **53.98** | **58.82** | 58.01 | 58.44 | 57.58 | 57.63 | 58.44 | 25.72 | 25.72 | **25.03** | 25.51 | 26.21 | 25.87 |
| 15 | b | 42.13 | 45.62 | 47.24 | 43.32 | 44.68 | **47.34** | 50.22 | 56.01 | 57.65 | 51.63 | 56.01 | **57.88** | 06.83 | 06.11 | 06.07 | 06.82 | 06.14 | **06.02** |
| | c | 36.82 | 37.64 | 36.33 | 35.61 | 37.02 | **37.73** | 36.22 | 36.94 | 34.85 | 34.47 | 36.72 | **37.23** | **18.52** | 18.91 | 19.12 | 19.13 | 18.91 | 18.72 |
| | e | 47.43 | 47.52 | 42.24 | 48.12 | 45.75 | **49.34** | 51.92 | 50.11 | 44.42 | 51.58 | 49.59 | **52.52** | 27.31 | 26.82 | 26.56 | 25.42 | 26.81 | **24.71** |
| | f | 60.72 | 62.41 | 61.01 | 59.22 | 61.73 | **63.98** | 70.24 | 72.93 | 69.14 | 68.42 | **73.13** | 72.84 | 25.72 | 23.87 | 26.82 | 26.73 | 23.71 | **23.62** |
| | s | 50.91 | 56.42 | 44.45 | 48.42 | 56.41 | **57.04** | 47.86 | 54.52 | 40.31 | 45.15 | 54.98 | **55.83** | 14.72 | 13.61 | 16.02 | 14.47 | 13.42 | **13.36** |
| | y | 47.32 | 50.34 | 49.02 | 49.94 | 50.98 | **51.64** | 54.23 | 57.91 | 58.27 | 57.31 | **58.82** | 58.31 | 25.47 | 25.76 | **24.62** | 25.34 | 25.52 | 25.67 |
| 20 | b | 42.62 | 44.24 | 44.38 | 43.27 | 43.93 | **45.14** | 52.07 | 52.93 | 54.32 | 51.74 | **55.18** | 54.72 | 06.34 | 06.12 | **05.74** | 06.34 | 05.92 | 05.94 |
| | c | 35.72 | 37.34 | 34.92 | 35.34 | 37.25 | **37.53** | 34.92 | 36.48 | 34.45 | 34.62 | 36.35 | **36.52** | 18.24 | 18.46 | **18.14** | 18.17 | 18.32 | 18.36 |
| | e | 46.75 | 49.13 | 40.01 | 46.85 | 47.69 | **49.68** | 48.54 | 51.41 | 39.54 | 51.15 | 49.05 | **52.37** | 27.23 | 26.32 | 27.14 | **25.34** | 27.32 | 26.04 |
| | f | 58.62 | 60.54 | 56.72 | 62.18 | 60.48 | **64.54** | 67.82 | 70.02 | 66.97 | 69.11 | **70.44** | 69.93 | 27.62 | 26.23 | 27.48 | 25.13 | 25.52 | **24.27** |
| | s | 46.67 | 56.72 | 42.96 | 47.32 | 56.03 | **57.54** | 42.73 | 55.52 | 38.31 | 43.92 | 55.88 | **56.47** | 15.44 | 13.26 | 16.01 | 14.12 | 13.15 | **12.62** |
| | y | 49.07 | 51.62 | 48.57 | 49.16 | 50.82 | **51.84** | 57.13 | 57.91 | 56.78 | 57.12 | 58.46 | **59.44** | 25.65 | 24.72 | **24.54** | 24.82 | 25.21 | 24.88 |
| 25 | b | 43.62 | 41.95 | 43.56 | 40.18 | 41.81 | **44.98** | 54.43 | 54.34 | 55.82 | 52.53 | 55.78 | **58.04** | 05.62 | 05.74 | 05.56 | 05.78 | 05.92 | **05.42** |
| | c | 36.02 | 36.92 | 36.63 | 36.18 | 37.55 | **37.79** | 37.32 | 37.93 | 34.88 | 35.04 | 38.16 | **38.43** | 16.62 | 17.58 | 17.62 | 17.51 | 17.34 | **17.35** |
| | e | 37.34 | 46.42 | 39.28 | 46.32 | 45.17 | **47.79** | 35.23 | 46.87 | 37.72 | 47.64 | 47.25 | **49.48** | 27.23 | 27.82 | 27.01 | **25.34** | 28.23 | 25.28 |
| | f | 59.64 | 59.78 | 62.03 | 59.82 | 59.64 | **63.63** | 68.11 | 70.02 | 68.18 | 68.53 | **70.35** | 68.81 | 28.92 | 26.23 | 28.28 | 27.49 | 27.32 | 27.73 |
| | s | 47.34 | 55.52 | 41.28 | 46.72 | 55.56 | **55.73** | 44.01 | **55.02** | 38.22 | 44.19 | 54.92 | 55.06 | 14.72 | **13.26** | 15.32 | 13.64 | 13.38 | 13.39 |
| | y | 49.12 | 51.54 | 48.92 | 49.13 | **51.52** | 51.15 | 57.92 | 60.03 | 57.18 | 57.42 | 59.92 | **60.21** | **23.88** | 24.33 | 24.22 | 24.01 | 24.38 | 24.13 |
| 30 | b | 38.35 | 41.72 | 40.84 | 40.48 | 40.52 | **42.93** | 50.44 | 52.92 | 51.75 | 48.73 | 51.21 | **51.68** | 06.22 | 06.28 | 05.73 | 06.24 | 06.58 | **05.43** |
| | c | 35.67 | 36.68 | 36.13 | 35.54 | 36.26 | **36.88** | 37.55 | 37.74 | 34.98 | 34.32 | 37.05 | **37.78** | **16.16** | 17.04 | 16.46 | 16.87 | 17.22 | 17.06 |
| | e | 43.54 | 46.78 | 40.64 | 49.26 | 48.12 | **50.04** | 46.26 | 50.02 | 38.34 | **53.35** | 52.88 | 51.42 | 27.05 | 26.92 | 26.36 | **24.34** | 25.88 | 25.42 |
| | f | 61.14 | 65.26 | 63.54 | 63.02 | 64.12 | **66.24** | 71.45 | 74.88 | 72.73 | 74.34 | 74.15 | **75.48** | 25.72 | 22.76 | 24.94 | 23.96 | 24.02 | **22.24** |
| | s | 44.46 | 54.92 | 38.34 | 43.68 | 55.44 | **56.45** | 42.03 | 55.12 | 33.86 | 41.32 | 55.04 | **56.25** | 15.48 | 12.92 | 16.26 | 14.12 | 13.07 | **12.75** |
| | y | 49.12 | 52.04 | 48.68 | 50.76 | 52.62 | **52.87** | 59.92 | **61.84** | 59.86 | 60.11 | 61.52 | 61.55 | 23.86 | 23.42 | 22.44 | **22.02** | 23.36 | 23.24 |
| Avg rank | - | 5 | 2 | 6 | 4 | 3 | **1** | 4 | 3 | 6 | 5 | 2 | **1** | 6 | 2 | 5 | 3 | 4 | **1** |

[1] "%" refers to the percentage of missing data analyzed (5%, 10%, 15%, 20%, 25%, and 30%).

[2] "Db" refers to the datasets used in the experimental setup, and these letters' abbreviations can be found in Table 3.

[3] Acronyms are related to each data imputation method tested, listed in S1 Table. Abbreviations.

### Multi-Label k-Nearest Neighbors

The results obtained with the ML-KNN classifier is shown in Table 7. As can be seen, EvoImp showed similar performance to the previous scenarios considering the BR and the HOMER classifiers. For instance, considering the primary analyzed metric (EM), EvoImp outperformed the baseline methods at 97.22%. Considering the ACC and HL, the EvoImp presented superior performance for 20 (55.55%) and 22 (61.11%) datasets, respectively.

### Classifier Chains

The results for the Classifier Chains are presented in Table 8. Again, EvoImp outperformed the baseline methods for all evaluation measures considered: EM with superiority in 32 out of 36 datasets (88.88%), ACC with 30 (83.33%), and HL with 22 (61.11%).

### Ensembles of Classifier Chains

The last scenario analyzed was considering the Ensemble Classifier Chains method. The results are shown in Table 9. The results obtained with the ECC (Table 9) also show a significant advantage of EvoImp over competitors in the analyses performed. However, EvoImp had the lowest performance, with numerical superiority in 29 (80.55%) datasets in the evaluation with EM, 16 (44.44%) for ACC, and 17 (47.22%) for HL.

In summary, the EvoImp performance for the ECC presents the same pattern described in the previous scenarios, demonstrating the EvoImp robustness.

## Discussion

In summary, EvoImp proved to be competitive in all classification scenarios, which underlines the fact that the optimization of imputation through evolutionary strategies, such as genetic algorithms, is an excellent alternative for handling missing values in the preprocessing phase of data analysis. It should be noted that the algorithm created performed optimizations based on simple imputation methods (applied to the initial population of EvoImp). Considering the computational experiments, other factors should be highlighted regarding the EvoImp performance:

- **Maximizing the labels' success**: The primary purpose of classification, particularly in this study, is the correct labeling of data instances, a task that is becoming increasingly complex in the multi-labeling scenario. In the EM measure, where the classifier must label all the classes of an instance correctly so that they can be counted correctly, the proposed method achieved better performance in 92.22% of all the datasets in all the scenarios. This performance is more evident in BR, HOMER, and ML-KNN, with 35 out of the 36 datasets. Another measure that allows this conclusion is ACC. The superior performance achieved by the EvoImp is more apparent in the analyses with the CC and HOMER classifiers (with 30 and 23 datasets, respectively). In general terms, EvoImp was better in 68.3% of all used datasets. This can be explained by the fact that this measure is flexible regarding the number of successes achieved by labels. For example, if an instance belongs to five labels and obtains four correct labels, it achieves an 80% degree of accuracy. At the same time, the excellent performance of ACC indicates that the classifier can increase its labeling capacity. This can be confirmed by analyzing the classification error evaluated using HL. In this metric, the proposed method obtained the lowest error (53.33%). It is worth mentioning that the results obtained reflect the lexicographic order chosen (as explained in subsection "Fitness function"), demonstrating the method's superiority over all the others. A comparison shows that when ACC increases, there is an automatic reduction in the HL error, justifying the usage of

**Table 7. Experimental results for the ML-KNN Classifier.**

| %[1] | Db[2] | Exact Match (↑) | | | | | | Accuracy (↑) | | | | | | Hamming Loss (↓) | | | | | |
|---|---|---|---|---|---|---|---|---|---|---|---|---|---|---|---|---|---|---|---|
| | | KMI[3] | KNNI[3] | MC[3] | CMC[3] | WKNNI[3] | EvoImp | KMI | KNNI | MC | CMC | WKNNI | EvoImp | KMI | KNNI | MC | CMC | WKNNI | EvoImp |
| 5 | b | 46.52 | 50.44 | 47.14 | 49.82 | 49.04 | **51.43** | 60.32 | 59.74 | 58.76 | 60.52 | 59.67 | **60.76** | 04.62 | 04.74 | 04.76 | **04.52** | 04.74 | 04.78 |
| | c | 36.06 | 35.62 | 36.24 | 36.46 | 36.08 | **36.64** | 61.22 | 61.40 | 61.59 | 62.26 | 60.61 | **62.28** | 13.42 | 13.54 | **13.11** | 13.23 | 13.52 | 13.24 |
| | e | 57.64 | 57.23 | 54.64 | 56.93 | 57.74 | **59.32** | 67.71 | 69.43 | 64.18 | 67.11 | **69.52** | 69.43 | 19.11 | 18.72 | 19.73 | 19.25 | 18.97 | **18.35** |
| | f | 61.42 | 61.61 | 61.25 | 60.98 | 61.82 | **62.44** | 70.63 | 70.91 | **73.26** | 72.42 | 71.21 | 71.54 | 27.72 | 27.31 | **26.44** | 26.72 | 27.38 | 27.13 |
| | s | 63.74 | 67.82 | 58.57 | 64.02 | 68.08 | **68.13** | 65.67 | 71.72 | 57.88 | 66.26 | **72.15** | **72.15** | 09.13 | 07.94 | 10.32 | 08.84 | **07.92** | **07.92** |
| | y | 56.24 | 57.88 | 55.96 | 55.62 | 58.03 | **58.21** | 72.46 | 72.92 | **73.16** | 72.85 | 72.66 | 72.62 | 19.21 | **18.78** | 18.82 | 18.94 | **18.78** | **18.78** |
| 10 | b | 47.02 | 46.04 | 44.28 | 43.93 | 46.07 | **47.92** | 55.94 | **57.62** | 55.53 | 56.24 | 56.88 | 57.04 | 04.62 | 04.64 | 04.66 | **04.51** | 04.68 | 04.62 |
| | c | 36.14 | 36.78 | 37.66 | 36.92 | 36.64 | **37.72** | 61.51 | 61.88 | 65.53 | 65.44 | 61.89 | **65.66** | 13.01 | 13.22 | **12.46** | 12.52 | 13.21 | **12.46** |
| | e | 55.46 | 55.18 | 50.82 | 56.06 | 54.07 | **56.96** | 65.62 | 64.05 | 56.69 | 65.92 | 64.06 | **66.67** | 20.26 | 20.62 | 21.87 | 19.72 | 20.96 | **19.62** |
| | f | 59.94 | 60.26 | 59.42 | 62.37 | 59.48 | **63.11** | 72.85 | 72.48 | 70.82 | 73.64 | 72.05 | **73.86** | 26.42 | 26.78 | 27.61 | 25.12 | 27.04 | **24.92** |
| | s | 60.92 | 69.46 | 48.92 | 59.88 | 69.72 | **69.87** | 62.11 | 74.82 | 43.89 | 61.34 | **74.96** | **74.96** | 09.52 | 07.38 | 11.92 | 09.56 | **07.32** | **07.32** |
| | y | 56.23 | 58.14 | 54.29 | 54.92 | 58.14 | **58.46** | 73.02 | 73.24 | **73.38** | 73.02 | 73.16 | 73.22 | 18.78 | 18.22 | 18.78 | 18.46 | 18.24 | **18.12** |
| 15 | b | 45.62 | 45.34 | 45.46 | 46.01 | 43.54 | **48.16** | 56.24 | 57.58 | 55.24 | 57.26 | 56.34 | **60.36** | 04.51 | 04.42 | 04.54 | **04.36** | 04.52 | **04.36** |
| | c | 36.52 | 37.74 | **39.56** | 39.02 | 37.74 | **39.56** | 62.22 | 61.01 | 67.28 | 67.09 | 61.46 | **67.47** | 12.62 | 12.74 | **11.64** | 11.72 | 12.76 | **11.64** |
| | e | 55.02 | 54.64 | 47.48 | 53.29 | 54.24 | **55.31** | 64.44 | 65.45 | 54.78 | 64.49 | 65.01 | **65.86** | 20.62 | 19.88 | 21.12 | 19.91 | 20.26 | **19.52** |
| | f | 60.56 | 61.14 | 56.62 | 58.48 | 60.62 | **61.87** | **72.96** | 72.02 | 72.14 | 70.32 | 71.75 | 72.42 | **26.16** | 26.62 | 27.34 | 28.18 | 27.12 | 26.41 |
| | s | 59.56 | 70.24 | 42.32 | 57.51 | **70.56** | **70.56** | 59.82 | 75.14 | 35.72 | 56.62 | **75.38** | 75.28 | 09.92 | **07.11** | 12.72 | 09.87 | **07.11** | **07.11** |
| | y | 53.74 | 59.82 | 55.81 | 55.66 | 59.62 | **59.92** | 69.65 | 73.19 | **73.81** | 73.02 | 73.02 | 73.27 | 19.29 | 18.22 | 17.92 | 17.87 | 17.56 | **17.44** |
| 20 | b | 43.82 | 45.62 | 45.04 | 46.49 | 44.52 | **50.91** | 57.56 | 57.14 | 54.97 | 55.62 | 56.65 | **57.64** | **04.14** | **04.14** | 04.22 | **04.14** | **04.14** | 04.71 |
| | c | 38.29 | 37.82 | **40.14** | 40.02 | 37.96 | **40.14** | 63.76 | 62.92 | 69.94 | **70.08** | 62.94 | 69.96 | 11.81 | 12.44 | **11.05** | **11.05** | 12.41 | **11.05** |
| | e | 53.42 | 55.14 | 41.06 | 52.07 | 55.26 | **55.96** | 59.41 | 63.97 | 42.25 | 61.26 | 64.84 | **65.52** | 21.21 | 20.18 | 22.79 | **19.32** | 20.31 | 20.26 |
| | f | 61.71 | 61.67 | 59.49 | 60.62 | 61.82 | **63.17** | 74.36 | 75.34 | 74.82 | 74.76 | 74.87 | **76.26** | 24.15 | 24.62 | 25.84 | 25.76 | 24.74 | **23.92** |
| | s | 54.15 | 71.92 | 39.88 | 54.59 | 71.41 | **72.06** | 51.82 | 77.34 | 34.32 | 52.56 | 77.22 | **77.26** | 10.31 | **06.72** | 12.56 | 10.01 | **06.72** | **06.72** |
| | y | 54.66 | 60.24 | 53.32 | 53.84 | 60.62 | **60.76** | 68.82 | 74.74 | **76.06** | 75.92 | 74.94 | 75.06 | 19.88 | 16.82 | 17.76 | 17.51 | **17.44** | **17.44** |
| 25 | b | 44.16 | 44.24 | 41.36 | 42.21 | 47.06 | **47.37** | 56.72 | 58.22 | 56.86 | 56.81 | 58.56 | **58.82** | 03.82 | 03.86 | 03.84 | 03.85 | **03.82** | **03.82** |
| | c | 39.28 | 39.16 | **41.94** | 41.25 | 38.79 | **41.94** | 63.32 | 62.92 | 70.64 | 69.62 | 62.66 | **70.71** | 11.52 | 12.16 | **10.34** | 10.55 | 12.18 | **10.34** |
| | e | 43.54 | 53.02 | 33.78 | 47.92 | 50.52 | **53.64** | 42.92 | 58.48 | 30.36 | 54.64 | 55.57 | **59.16** | 21.92 | 21.54 | 22.72 | **19.74** | 22.02 | 21.24 |
| | f | 60.56 | 60.32 | 62.24 | 58.76 | 59.84 | **62.55** | 75.42 | 75.29 | 74.21 | **75.82** | 74.95 | 74.36 | 26.39 | 25.02 | 25.96 | 26.52 | 25.84 | **25.64** |
| | s | 52.65 | 72.24 | 36.32 | 50.96 | 72.34 | **72.58** | 49.64 | 78.31 | 32.02 | 48.36 | 78.17 | **78.46** | 10.52 | **06.41** | 12.44 | 10.42 | **06.41** | **06.41** |
| | y | 54.42 | 61.25 | 54.49 | 54.31 | 61.06 | **61.88** | 76.76 | 74.91 | 77.14 | **77.36** | 75.05 | 75.21 | 17.02 | **16.16** | 16.82 | 16.91 | **16.16** | 16.78 |
| 30 | b | 42.32 | 42.16 | 38.58 | 42.57 | 39.82 | **46.99** | 55.11 | 54.46 | 55.34 | 55.26 | 54.25 | **55.76** | **03.92** | 04.01 | **03.92** | **03.92** | 04.08 | 04.06 |
| | c | 39.74 | 38.26 | 42.52 | 42.64 | 38.12 | **42.71** | 65.76 | 63.84 | 74.08 | **73.65** | 64.24 | **73.65** | 10.92 | 11.86 | **09.68** | 09.72 | 11.74 | 09.78 |
| | e | 50.02 | 52.99 | 31.62 | 52.64 | 54.94 | **55.08** | 57.56 | 63.01 | 28.56 | 64.88 | 64.86 | **65.57** | 20.91 | 20.35 | 22.46 | 19.62 | 19.68 | **19.51** |
| | f | **65.72** | 63.36 | 61.38 | 64.32 | 60.71 | 65.19 | **79.12** | 77.45 | 81.27 | 77.99 | 77.32 | 76.06 | **22.61** | 24.22 | 23.45 | 22.98 | 25.16 | 24.12 |
| | s | 51.32 | 72.26 | 34.62 | 48.84 | 72.26 | **72.32** | 49.44 | 78.92 | 31.56 | 45.68 | **79.12** | 79.04 | 10.62 | 06.26 | 12.18 | 10.44 | **06.12** | **06.12** |
| | y | 52.56 | 62.58 | 53.72 | 54.31 | 62.88 | **63.09** | 73.61 | 75.85 | **79.42** | 79.24 | 76.16 | 76.12 | 18.88 | 15.26 | 16.12 | 16.14 | **15.14** | **15.14** |
| Avg rank | - | 5 | 2 | 6 | 4 | 3 | **1** | 6 | 3 | 5 | 2 | 4 | **1** | 5 | 2 | 6 | 3 | 4 | **1** |

[1] "%" refers to the percentage of missing data analyzed (5%, 10%, 15%, 20%, 25%, and 30%).

[2] "Db" refers to the datasets used in the experimental setup, and these letters' abbreviations can be found in Table 3.

[3] Acronyms are related to each data imputation method tested, listed in S1 Table. Abbreviations.

**Table 8. Experimental results for the Classifier Chains.**

| %¹ | Db² | Exact Match (↑) | | | | | | Accuracy (↑) | | | | | | Hamming Loss (↓) | | | | | |
|---|---|---|---|---|---|---|---|---|---|---|---|---|---|---|---|---|---|---|---|
| | | KMI³ | KNNI³ | MC³ | CMC³ | WKNNI³ | EvoImp | KMI | KNNI | MC | CMC | WKNNI | EvoImp | KMI | KNNI | MC | CMC | WKNNI | EvoImp |
| 5 | b | 49.62 | 48.56 | 49.91 | 50.45 | 50.21 | **51.46** | 61.52 | 60.81 | 61.44 | 60.99 | 61.62 | **62.26** | 05.21 | 05.25 | 05.06 | 05.11 | 05.36 | **05.02** |
| | c | 34.64 | 34.92 | 34.34 | 34.56 | 34.81 | **35.34** | 40.06 | 40.14 | 40.21 | **40.46** | 40.05 | 40.32 | 17.34 | 17.46 | 17.14 | **17.01** | 17.42 | 17.13 |
| | e | 49.82 | 49.91 | 45.86 | 48.82 | 48.81 | **52.06** | 55.45 | 55.68 | 52.82 | 53.59 | 54.76 | **56.56** | 25.72 | **25.42** | 27.34 | 26.72 | 25.86 | 25.61 |
| | f | 59.52 | 58.98 | 60.56 | 59.76 | 59.32 | **62.26** | 66.72 | 67.11 | 68.62 | 68.18 | 67.92 | **70.06** | 29.71 | 29.82 | 28.26 | 28.71 | 29.08 | **26.86** |
| | s | 56.52 | 59.78 | 51.51 | 54.26 | 58.17 | **59.96** | 56.82 | 61.84 | 51.42 | 55.46 | 60.36 | **62.02** | 14.14 | 13.62 | 15.52 | 13.88 | 14.04 | **13.52** |
| | y | 49.29 | 49.82 | 48.14 | 48.12 | 49.98 | **50.36** | 56.22 | 56.38 | 54.54 | 55.02 | 56.44 | **56.48** | 26.56 | 26.48 | 26.61 | 26.46 | **26.42** | 26.51 |
| 10 | b | 46.22 | 48.16 | 48.44 | 45.12 | 45.92 | **49.56** | 57.61 | 60.52 | 60.38 | 55.76 | 58.13 | **61.62** | 05.41 | 05.36 | 05.02 | 05.38 | 05.56 | **04.92** |
| | c | **36.18** | 35.56 | 34.82 | 35.08 | 35.31 | 36.06 | **45.45** | 42.02 | 40.76 | 40.61 | 41.68 | 41.82 | **16.32** | 16.52 | 16.31 | 16.36 | 16.72 | 16.56 |
| | e | 45.92 | 49.48 | 41.71 | 49.66 | 48.38 | **50.92** | 52.36 | 54.25 | 46.07 | 52.02 | **55.06** | 54.92 | 28.08 | 26.89 | 29.52 | 27.24 | 26.56 | **26.44** |
| | f | 57.16 | 59.62 | 57.74 | 58.33 | 61.05 | **61.58** | 69.72 | 71.76 | 71.41 | 68.12 | 72.26 | **72.82** | 28.18 | 27.12 | 26.76 | 28.81 | 26.08 | **25.36** |
| | s | 53.54 | 59.92 | 48.46 | 52.72 | 58.71 | **60.08** | 54.16 | 61.85 | 46.32 | 51.16 | 60.32 | **61.96** | 14.12 | 13.18 | 15.77 | 14.09 | 13.66 | **13.12** |
| | y | 48.52 | 49.26 | 47.24 | 46.96 | 49.32 | **50.56** | 55.34 | 55.69 | 53.82 | 53.89 | 56.92 | **57.75** | 26.24 | 26.62 | 26.54 | 26.26 | 26.28 | **25.66** |
| 15 | b | 44.52 | 45.38 | **46.62** | 41.91 | 44.28 | **46.62** | 55.16 | 56.21 | **57.62** | 52.88 | 55.62 | 57.61 | 05.72 | 05.32 | 05.08 | 05.59 | 05.32 | **05.06** |
| | c | **35.82** | 35.06 | 35.39 | 34.62 | 34.51 | 35.75 | 41.46 | 41.92 | 41.18 | 40.92 | 41.36 | **42.52** | 16.06 | 16.28 | **15.62** | 15.76 | 16.38 | 15.67 |
| | e | 47.96 | 47.92 | 42.16 | 50.31 | 47.88 | **50.96** | 50.92 | **56.28** | 42.06 | 55.24 | 53.76 | 56.08 | 27.12 | 26.16 | 27.42 | 24.54 | 26.76 | **24.18** |
| | f | 61.52 | 62.36 | 58.11 | 60.89 | 62.82 | **64.89** | 72.02 | 73.46 | 72.09 | 72.92 | 73.74 | **74.19** | 26.12 | 24.87 | 26.42 | 25.74 | 24.32 | **23.62** |
| | s | 51.72 | 59.86 | 47.31 | 50.26 | 59.02 | **60.18** | 52.04 | 62.42 | 44.26 | 48.61 | 61.02 | **62.75** | 14.56 | 12.92 | 15.24 | 13.56 | 13.12 | **12.82** |
| | y | 46.11 | 51.36 | 48.34 | 46.72 | 51.06 | **51.82** | 53.16 | 57.91 | 55.22 | 53.88 | 57.46 | **58.16** | 26.02 | 25.05 | 25.14 | 26.02 | 25.38 | **24.96** |
| 20 | b | 42.02 | 44.84 | 43.96 | 43.46 | 44.33 | **46.17** | 53.25 | 57.48 | 55.44 | 53.13 | 55.42 | **58.14** | 05.52 | 05.06 | 05.07 | 05.32 | 05.26 | **04.92** |
| | c | 35.74 | 36.16 | 34.91 | 34.78 | 35.26 | **36.62** | 41.26 | 42.34 | 41.85 | 40.38 | 41.34 | **43.48** | 15.46 | 15.82 | **14.84** | 15.21 | 16.16 | 15.32 |
| | e | 44.98 | 47.93 | 38.52 | 47.94 | 45.92 | **49.38** | 48.92 | 51.91 | 36.74 | 52.58 | 50.03 | **53.55** | 27.82 | 26.68 | 28.39 | 26.62 | 27.64 | **25.86** |
| | f | 57.62 | 61.38 | 58.97 | 60.76 | 59.02 | **61.88** | 68.91 | 70.86 | 70.72 | 70.54 | **71.26** | **71.26** | 28.92 | **25.51** | 26.75 | 26.62 | 25.66 | 25.57 |
| | s | 46.52 | **59.94** | 42.26 | 48.05 | 59.18 | 59.76 | 44.22 | 61.64 | 38.38 | 45.72 | **61.62** | **61.62** | 15.12 | **12.64** | 16.22 | 13.79 | 12.72 | 12.69 |
| | y | 48.82 | 49.91 | 46.06 | 45.92 | 51.38 | **51.41** | 56.42 | 56.56 | 53.08 | 53.16 | 58.51 | **58.68** | 25.66 | 25.88 | 25.62 | 25.64 | 24.82 | **24.71** |
| 25 | b | 44.13 | 43.42 | 43.15 | 41.28 | 45.42 | **45.78** | 54.62 | 55.51 | 55.06 | 51.62 | 58.34 | **58.42** | 05.06 | 05.21 | 04.75 | 04.92 | 04.74 | **04.72** |
| | c | **37.56** | 35.88 | 36.12 | 35.51 | 36.36 | 37.23 | 43.72 | 41.91 | 43.26 | 42.72 | 42.81 | **44.86** | 14.64 | 15.72 | **14.24** | 14.28 | 15.42 | 14.56 |
| | e | 42.24 | 46.12 | 37.26 | 45.08 | 46.22 | **48.76** | 43.02 | 49.81 | 35.92 | 48.76 | 49.31 | **52.19** | 26.82 | 27.18 | 27.26 | 27.61 | 27.15 | **26.22** |
| | f | 59.32 | 60.66 | 60.92 | 59.28 | 61.24 | **62.35** | **72.66** | 72.32 | 72.44 | 71.26 | 72.58 | 72.22 | 26.87 | 25.92 | 26.84 | 26.92 | 25.65 | **25.61** |
| | s | 46.75 | 59.94 | 39.62 | 45.96 | 60.12 | **61.56** | 43.94 | 63.26 | 36.72 | 42.92 | 62.26 | **63.98** | 14.65 | 12.34 | 15.82 | 13.76 | 12.52 | **11.77** |
| | y | 46.15 | 51.52 | 46.63 | 46.28 | 51.39 | **51.67** | 54.12 | 58.72 | 54.55 | 53.59 | 58.92 | **58.97** | 24.36 | 24.52 | 24.55 | 24.52 | 24.56 | **24.48** |
| 30 | b | 42.06 | 40.62 | 44.97 | 40.66 | 41.34 | **45.49** | 54.25 | 52.87 | 55.92 | 51.46 | 52.85 | **56.49** | 05.02 | 05.28 | 04.76 | 05.11 | 05.12 | **04.64** |
| | c | 36.83 | 35.05 | 35.42 | 36.58 | 34.92 | **38.76** | 44.24 | 41.92 | 43.76 | 44.25 | 40.92 | **45.81** | 14.02 | 15.26 | **13.43** | 13.49 | 15.44 | 13.82 |
| | e | 42.55 | 44.82 | 38.18 | 48.09 | 45.74 | **48.80** | 47.06 | 50.22 | 37.79 | 52.95 | 50.91 | **54.42** | 26.90 | 27.44 | 26.66 | 24.54 | 27.48 | **24.26** |
| | f | 61.60 | 61.65 | 59.86 | 62.92 | 63.31 | **64.14** | 75.42 | 73.26 | 75.18 | 74.82 | 74.19 | **75.47** | 24.62 | 26.06 | 24.98 | 24.76 | 25.12 | **24.14** |
| | s | 43.40 | 59.85 | 38.73 | 45.44 | 58.83 | **60.15** | 41.62 | 62.53 | 34.64 | 43.23 | 62.18 | **63.29** | 15.20 | **12.02** | 15.36 | 13.21 | 12.25 | 12.03 |
| | y | 49.77 | 52.86 | 47.40 | 47.27 | 52.56 | **53.12** | 57.91 | 60.48 | 54.96 | 55.64 | 60.32 | **60.65** | 24.10 | 23.23 | 23.35 | **23.01** | 23.46 | 23.12 |
| Avg rank | - | 4 | 2 | 6 | 5 | 3 | **1** | 4 | 2 | 5 | 6 | 3 | **1** | 6 | 3 | 4 | 2 | 5 | **1** |

¹ "%" refers to the percentage of missing data analyzed (5%, 10%, 15%, 20%, 25%, and 30%).

² "Db" refers to the datasets used in the experimental setup, and these letters' abbreviations can be found in Table 3.

³ Acronyms are related to each data imputation method tested, listed in S1 Table. Abbreviations.

**Table 9. Experimental Results for the Ensemble of Classifier Chains.**

| %[1] | Db[2] | Exact Match (↑) | | | | | | Accuracy (↑) | | | | | | Hamming Loss (↓) | | | | | |
|---|---|---|---|---|---|---|---|---|---|---|---|---|---|---|---|---|---|---|---|
| | | KMI[3] | KNNI[3] | MC[3] | CMC[3] | WKNNI[3] | EvoImp | KMI | KNNI | MC | CMC | WKNNI | EvoImp | KMI | KNNI | MC | CMC | WKNNI | EvoImp |
| 5 | b | 53.72 | 54.43 | 52.75 | 52.79 | 53.61 | **54.69** | 64.42 | 65.78 | 64.31 | 63.40 | **65.76** | 65.32 | 04.23 | 04.26 | 04.27 | **04.21** | 04.26 | 04.28 |
| | c | 54.52 | 54.88 | 54.29 | 54.12 | **55.16** | 36.64 | 54.36 | 55.02 | **56.89** | 55.72 | 54.64 | 56.09 | 14.12 | 14.16 | 13.79 | 13.82 | 14.16 | **13.72** |
| | e | 55.56 | 56.82 | 52.16 | 58.26 | 56.92 | **58.68** | 61.92 | **65.86** | 59.12 | 64.57 | 63.42 | 63.93 | 20.22 | 19.05 | 20.93 | **18.94** | 19.76 | 19.18 |
| | f | 61.32 | 62.09 | 61.60 | 62.62 | 62.25 | **65.68** | 71.02 | 70.26 | 70.72 | 71.58 | 69.72 | **72.56** | 25.71 | 25.46 | 26.22 | 25.67 | 25.92 | **24.84** |
| | s | 59.92 | 63.93 | 56.14 | 57.92 | **64.46** | **64.46** | 59.82 | 65.84 | 54.83 | 57.32 | 65.84 | **65.89** | 09.82 | **08.83** | 10.30 | 09.82 | 08.87 | 08.88 |
| | y | 54.52 | 54.86 | 54.24 | 54.12 | 55.18 | **55.39** | 69.02 | 68.80 | **69.44** | 69.41 | 69.13 | 69.39 | 20.52 | 20.46 | 19.91 | **19.82** | 20.36 | 20.23 |
| 10 | b | 45.25 | 46.62 | 49.13 | 47.99 | 48.10 | **49.28** | 60.02 | **61.97** | 61.32 | 58.92 | 61.65 | 61.26 | 04.42 | 04.41 | **04.23** | 04.35 | 04.36 | 04.27 |
| | c | 54.16 | **55.82** | 51.96 | 52.98 | 55.72 | 37.51 | 56.16 | 55.54 | 57.22 | 57.58 | 55.46 | **58.14** | 13.55 | 13.83 | **13.14** | 13.18 | 13.86 | 13.33 |
| | e | 53.82 | 53.87 | 46.85 | 54.20 | 54.51 | **55.35** | 60.83 | 60.71 | 52.02 | **61.93** | 60.62 | 61.14 | 21.23 | **20.92** | 22.46 | 20.32 | 21.27 | **20.92** |
| | f | 61.33 | 61.37 | 61.52 | 62.78 | 60.93 | **65.03** | 71.51 | 72.28 | 72.92 | 73.33 | 73.09 | **74.05** | 26.24 | 25.52 | **24.46** | 24.84 | 25.82 | 24.83 |
| | s | 59.05 | 64.82 | 52.73 | 56.77 | **65.26** | **65.26** | 58.14 | 66.28 | 48.72 | 55.16 | **66.87** | **66.87** | 09.82 | 08.74 | 10.93 | 09.78 | **08.42** | **08.42** |
| | y | 54.19 | 55.82 | 51.97 | 52.93 | 54.08 | **56.06** | 70.07 | 70.32 | 69.49 | 69.72 | 70.17 | **70.66** | 19.92 | 19.78 | 19.91 | 19.82 | 19.79 | **19.63** |
| 15 | b | 47.24 | 47.39 | 47.82 | 44.18 | 47.33 | **50.04** | 58.07 | **61.09** | 60.81 | 57.66 | 60.48 | 59.64 | 04.41 | **04.23** | **04.23** | 04.37 | 04.28 | 04.25 |
| | c | 51.92 | **56.09** | 52.90 | 53.77 | 56.06 | 38.25 | 57.78 | 57.62 | **60.23** | 58.54 | 57.93 | 60.12 | 12.96 | 13.25 | **12.37** | 12.53 | 13.12 | 12.39 |
| | e | 53.02 | 52.78 | 45.13 | 54.54 | 52.02 | **55.34** | 59.75 | **63.78** | 48.52 | 62.46 | 61.01 | 62.49 | 21.35 | 20.03 | 22.67 | 19.68 | 20.72 | **19.29** |
| | f | 62.52 | 63.89 | 59.67 | 60.45 | 64.32 | **67.09** | 73.74 | 73.05 | 71.20 | 71.36 | 73.72 | **74.48** | 24.34 | 23.95 | 26.57 | 26.02 | 23.69 | **22.92** |
| | s | 56.64 | 64.82 | 49.73 | 53.68 | 64.62 | **64.88** | 56.32 | 65.56 | 46.72 | 51.20 | **66.06** | 65.52 | 10.17 | **08.52** | 10.96 | 10.04 | 08.69 | **08.52** |
| | y | 51.94 | 56.01 | 52.92 | 53.78 | 56.09 | **56.32** | 66.46 | 70.52 | 71.58 | 71.05 | 70.06 | **72.08** | 20.02 | 19.39 | 19.91 | 18.87 | 19.39 | **19.16** |
| 20 | b | 47.15 | 46.53 | 46.58 | 46.92 | 47.59 | **51.07** | 56.92 | 59.26 | 58.68 | 58.22 | 58.47 | **59.46** | 04.15 | 04.16 | 04.02 | 04.08 | 04.19 | **03.96** |
| | c | 53.82 | 56.36 | 52.78 | 52.46 | **57.17** | 39.12 | 58.82 | 57.67 | 61.92 | 62.09 | 57.35 | **62.51** | 12.26 | 12.88 | 11.66 | 11.64 | 12.98 | **11.62** |
| | e | 51.32 | 53.28 | 45.56 | 51.98 | 52.40 | **54.67** | 57.16 | 60.01 | 43.78 | 59.32 | 60.92 | **61.16** | 20.74 | 20.42 | 22.29 | **19.44** | 20.82 | 20.07 |
| | f | 59.92 | 62.56 | 61.07 | 61.74 | 62.02 | **63.89** | 71.45 | 72.49 | 71.63 | 70.67 | 72.42 | **73.46** | 26.57 | 24.09 | 25.55 | 26.79 | 24.34 | **23.34** |
| | s | 49.85 | 65.32 | 46.39 | 53.87 | 65.26 | **65.37** | 47.12 | 66.96 | 41.75 | 51.90 | 66.88 | **66.95** | 11.12 | **08.23** | 11.54 | 09.56 | 08.35 | **08.23** |
| | y | 53.88 | 56.32 | 52.70 | 52.45 | 57.19 | **57.42** | 69.36 | 70.97 | **72.62** | 72.06 | 70.97 | 71.24 | 19.82 | 19.09 | **18.52** | 18.67 | 18.92 | 18.80 |
| 25 | b | 44.15 | 43.63 | 45.42 | 45.77 | 44.26 | **48.56** | 59.72 | **59.82** | 58.79 | 58.36 | 59.54 | 59.55 | 03.72 | 03.77 | 03.76 | 03.75 | 03.82 | **03.65** |
| | c | 52.92 | **57.29** | 53.43 | 53.14 | 57.16 | 40.32 | 61.26 | 59.18 | **64.93** | 63.64 | 59.72 | 63.56 | 11.68 | 12.46 | 10.92 | **10.14** | 12.46 | 11.24 |
| | e | 44.38 | 51.93 | 38.00 | 50.97 | 50.42 | **53.83** | 46.62 | 56.54 | 38.08 | 58.62 | 55.46 | **61.24** | 22.02 | 22.06 | 22.28 | 18.42 | 22.37 | **18.24** |
| | f | 61.54 | 61.90 | 61.98 | 60.86 | 60.82 | **63.44** | 73.02 | **74.30** | 73.56 | 73.24 | 72.38 | 73.11 | 25.88 | **24.02** | 25.56 | 25.14 | 25.21 | 25.48 |
| | s | 51.36 | 66.55 | 44.27 | 51.42 | 65.76 | **66.67** | 48.65 | 68.52 | 40.28 | 50.07 | 68.02 | **68.79** | 10.52 | **07.94** | 11.56 | 09.62 | 08.17 | 07.99 |
| | y | 52.92 | 57.28 | 53.44 | 53.15 | 57.16 | **57.72** | 73.07 | 71.45 | **73.42** | 72.84 | 71.96 | 72.02 | 17.88 | 18.74 | **17.73** | 18.05 | 18.42 | 18.27 |
| 30 | b | 42.35 | 42.28 | 46.43 | 47.35 | 42.02 | **49.86** | 55.14 | 54.92 | 54.94 | **58.26** | 55.07 | 56.51 | 03.95 | 04.17 | 03.92 | **03.86** | 03.82 | 03.88 |
| | c | 54.82 | **58.26** | 53.42 | 53.47 | 57.89 | 40.80 | 63.32 | 58.67 | **66.69** | 66.12 | 59.77 | 66.26 | 11.02 | 12.28 | 10.24 | **10.32** | 12.11 | 10.39 |
| | e | 49.34 | 53.72 | 35.23 | 54.67 | 54.15 | **55.16** | 56.01 | 62.98 | 35.36 | **65.25** | 63.38 | 64.67 | 21.42 | 19.72 | 22.10 | 18.55 | 19.76 | **18.22** |
| | f | **65.73** | 63.65 | 62.47 | 64.82 | 62.89 | 65.64 | **78.23** | 75.35 | 76.48 | 76.72 | 74.86 | 77.47 | 22.76 | 23.27 | 24.26 | 22.82 | 24.07 | **22.52** |
| | s | 48.96 | 65.70 | 42.77 | 50.54 | 65.02 | **65.76** | 47.37 | 68.92 | 38.75 | 48.69 | 67.52 | **68.97** | 10.92 | **07.65** | 11.32 | 09.67 | 08.02 | **07.65** |
| | y | 54.82 | 58.28 | 53.49 | 53.42 | 57.87 | **58.34** | 72.92 | 73.19 | 74.92 | **75.28** | 73.42 | 73.46 | 18.28 | 17.71 | 16.92 | **16.87** | 17.76 | 17.62 |
| Avg rank | - | 5 | 2 | 6 | 4 | 3 | **1** | 6 | 2 | 5 | 3 | 4 | **1** | 6 | 3 | 4 | 2 | 5 | **1** |

[1] "%" refers to the percentage of missing data analyzed (5%, 10%, 15%, 20%, 25%, and 30%).

[2] "Db" refers to the datasets used in the experimental setup, and these letters' abbreviations can be found in Table 3.

[3] Acronyms are related to each data imputation method tested, listed in S1 Table. Abbreviations.

lexicographical order instead of more complex approaches, such as Pareto Frontier Analysis, used to deal with conflicting measures.

- **Superior performance in datasets over different domains and sizes**: The six datasets used in the experiments can be divided in terms of i) different domains—the multi-label datasets used were related to the areas of audio (1), music (2), image (2), and biology (1); ii) their sizes—considering the number of instances and attributes, as was done by [54]. These datasets were curated to provide a robust experimental setup, simulating diverse real-world problems. It was noted that EvoImp performed superior in all the tests, proving that the method is robust on datasets of different domains and sizes.

- **Stable performance in the distribution rates of the missing values under study**: A critical evaluation of this study is related to the relationship between the missing values percentage and the performance measures. The results show that the EvoImp maintains its consistency, even with variations, which, in this study, was between 5% to 30% (with a rate of k = 5%). These rates agree with those used in most studies in the literature—one related work that addresses this discussion is [17]. A total of 48 related articles from 2011 to 2021 were selected in this investigation. About missing rates, this review indicated that 60,4% used missing rates $< = 30\%$ or did not reveal their missing rates for the experimentation.

The above aspects demonstrate that EvoImp is suitable for missing value treatments in real-world scenarios.

## Conclusion and suggestions for future work

The data analyses conducted in real-world datasets make it clear that there is a critical need to handle missing values in multi-label classification domain. The ubiquitous presence of MVs and the fact that most of the techniques employed only work or ensure good performance when applied to datasets with complete cases underlines the need to tackle this problem. Data imputation methods have emerged as an alternative solution, searching for plausible values to fill the missing ones.

Therefore, we proposed in this study the EvoImp, an imputation method based on genetic algorithms for the optimization of multiple imputations for missing data applied to multi-label learning. For validation, the method was submitted to an extensive experimental benchmarking process with various multi-label datasets and compared with other state-of-the-art imputation methods. Six missing value rates are applied to the datasets to simulate the MCAR mechanism. The results were analyzed using five classifiers: Binary Relevance, Hierarchy of Multi-label Classifier, Multi-Label k-Nearest Neighbors, Classifier Chains, and Ensembles of Classifier Chains. Three well-known evaluation measures were adopted to assess the experiments: Exact Match, Accuracy, and Hamming-loss.

EvoImp achieved exceptional results in all the scenarios evaluated, being quantitatively superior to the others. These outstanding results make it possible to conclude that the proposed method is suitable for application in real-world scenarios. In addition to a novel approach for dealing with MV in multi-label classification, the present works contribute to the body of knowledge by: i) assessing the impact of missing data on multi-label classification to improve classification robustness; ii) providing an extensive experimental comparison of many state-of-the-art data imputation algorithms, multi-label machine learning classifiers, and performance measures; iii) making source codes and experiments results in a GitHub repository.

In future work, we want to evaluate other missingness mechanisms apart from MCAR and adjust the method for handling high rates of missing data ($> 30\%$). Experiments could also be

performed to make the EvoImp learn its parameters (AutoML). Finally, we would like to investigate the Influence of Cardinality and Density Characteristics on Multi-Label Learning with missing values.

## Supporting information

**S1 Table. Abbreviations.**
(PDF)

## Author Contributions

**Conceptualization:** Antonio Fernando Lavareda Jacob Junior, Adamo Lima de Santana, Fabio Manoel Franca Lobato.

**Data curation:** Fabio Manoel Franca Lobato.

**Formal analysis:** Fabio Manoel Franca Lobato.

**Investigation:** Antonio Fernando Lavareda Jacob Junior.

**Methodology:** Fabio Manoel Franca Lobato.

**Software:** Fabricio Almeida do Carmo.

**Supervision:** Ewaldo Eder Carvalho Santana, Fabio Manoel Franca Lobato.

**Validation:** Antonio Fernando Lavareda Jacob Junior, Fabio Manoel Franca Lobato.

**Visualization:** Antonio Fernando Lavareda Jacob Junior.

**Writing – original draft:** Antonio Fernando Lavareda Jacob Junior.

**Writing – review & editing:** Antonio Fernando Lavareda Jacob Junior, Fabricio Almeida do Carmo, Adamo Lima de Santana, Ewaldo Eder Carvalho Santana, Fabio Manoel Franca Lobato.

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
