## [Decision Letter · Decision Letter 0]

10 Jul 2023

PONE-D-23-16570GAMultImp: Multiple Imputation of Multi-label classification data With a genetic algorithmPLOS ONE

Dear Dr. JACOB JUNIOR,

Thank you for submitting your manuscript to PLOS ONE. After careful consideration, we feel that it has merit but does not fully meet PLOS ONE’s publication criteria as it currently stands. Therefore, we invite you to submit a revised version of the manuscript that addresses the points raised during the review process.

We look forward to receiving your revised manuscript.

Kind regards,

Abdullah Hussein Abdullah Alamoodi, Ph.D.

Academic Editor

PLOS ONE

Journal Requirements:

"FMFL was financed in part by the National Council for Scientific and Technological Development (CNPq, Brazil) under Grant

147336/2020-1."

4. Thank you for stating the following in the Acknowledgments Section of your manuscript: "This study was financed in part by the National Council for Scientific and Technological 496

Development (CNPq, Brazil) under Grant 147336/2020-1"

"FMFL was financed in part by the National Council for Scientific and Technological Development (CNPq, Brazil) under Grant

147336/2020-1."

Reviewers' comments:

Reviewer's Responses to Questions

**Comments to the Author**

1. Is the manuscript technically sound, and do the data support the conclusions?

Reviewer #1: Yes

Reviewer #2: Yes

2. Has the statistical analysis been performed appropriately and rigorously? 

Reviewer #1: Yes

Reviewer #2: I Don't Know

3. Have the authors made all data underlying the findings in their manuscript fully available?

Reviewer #1: Yes

Reviewer #2: Yes

4. Is the manuscript presented in an intelligible fashion and written in standard English?

Reviewer #1: Yes

Reviewer #2: No

5. Review Comments to the Author

Reviewer #1: The authors describe Multiple Imputation of Multi-label classification data with a genetic algorithm, which is a very interesting study. The authors propose a novel imputation method based on genetic algorithm for optimizing multiple data imputations, and applies the proposed method in multi-label learning and evaluated its performance using six synthetic databases, considering various missing values distribution scenarios. But there is still room for improvement in the article.

1. Abstract: "Based on genetic algorithm" is too general to reflect the basic principles of the proposed method.

2. Introduction: line16 The abbreviation MLOPs appears for the first time, please provide their full names

3. In addition to the differences between the two classification tasks, I believe the author also needs to clarify the differences between single label data and multi label data, as well as why MultImp cannot or is not suitable for applying to multi label data, and then propose their own GAMultImp.

4. GAMultImp - Proposed Method

It is necessary to clarify whether the determination of the parameters of the crossover operator in line236 was determined through extensive experiments conducted in this article, or was it based on the results of experiments conducted by others.

Is the candidate value exchanged by the line240 mutation operator computationally generated, randomly generated, or self-defined. Exact explanation is required.

above all, I think the paper needs to be major revised.

Reviewer #2: This manuscript uses genetic algorithms to solve the multiple imputations problem in multi-label learning. The experiments are adequate and the methods are analyzed and discussed. However, the manuscript still suffers from the following problems.

1. The abstract is too verbose about the background and the paper’s focus and innovation are unclear.

2. The first two introductory paragraphs in Introduction Section are confusing and it is impossible to understand what the author is trying to convey.

3. Please redraw the Algorithm 1 flowchart.

4. What do the ‘16 scenario databases’ mentioned in Binary Relevance in Results and Analysis mean? If it refers to the number of datasets that performed well in terms of accuracy, the results shown in Table 3 should be 18.

6. PLOS authors have the option to publish the peer review history of their article (what does this mean?). If published, this will include your full peer review and any attached files.

Reviewer #1: No

Reviewer #2: No

---

## [Author Response · Author response to Decision Letter 0]

2 Aug 2023

We are grateful for the opportunity to revise and resubmit our paper. We have made sincere efforts to address all of the reviewers’ remarks. 

In the subsequent pages, we present detailed responses to the comments provided by the two reviewers. The original comments are highlighted in bold, while our responses are presented in standard text.

Additionally, we provide a summary of the four major revisions undertaken below.

Abstract and Introduction Section. We performed the reduction of problem contextualization and emphasized the contribution of the work, as well as highlighted that the data and source codes used are available in a Github repository, following the principles of open science. Please also see our responses to Reviewer 1, point #1 and Reviewer 2, points #1 and #2, for more details.

Change in the method's name and paper’s title. In order to better clarify the difference between the method presented in this article and the strategy on which this method was based (MultImp), we opted for a change in the nomenclature of the method. In this case, we modified from GAMultImp to EvoImp, to reference the terms evolutionary and imputation. As a consequence, the article's title was adjusted accordingly. For more details, see our responses to Reviewer 1, point #3.

Proposed Method Section. We clarified how the parameters used in the crossover and mutation operators were determined. Additionally, a review of the method's algorithm was carried out. Please see our responses to Reviewer 1, point #3 and Reviewer 2, point #3. for details.

Results and Analysis Section. We reviewed all the values of the method's results obtained in the experiments, for which we appreciate the careful attention of Reviewer 2. For more details, please see our responses to Reviewer 2, point #4.

Thank you again for offering us this valuable opportunity to revise and resubmit our manuscript. We hope you find the quality of this paper has been improved.

Kind regards,

The Authors

---

## [Decision Letter · Decision Letter 1]

3 Oct 2023

PONE-D-23-16570R1EvoImp: Multiple Imputation of Multi-label classification data With a genetic algorithmPLOS ONE

Dear Dr. JACOB JUNIOR,

Thank you for submitting your manuscript to PLOS ONE. After careful consideration, we feel that it has merit but does not fully meet PLOS ONE’s publication criteria as it currently stands. Therefore, we invite you to submit a revised version of the manuscript that addresses the points raised during the review process. We strongly recommend to utilize professional English Writing Service to improve the manuscript organization and readability.

We look forward to receiving your revised manuscript.

Kind regards,

Mohammad A. Al-Mamun, PhD

Academic Editor

PLOS ONE

Additional Editor Comments 

This is an important study for addressing the imputation of multiclass classification problem. I suggest to use a English Writing service to improve the manuscript readability. So, I suggest a thorough review and rewrite of all sections. My major comments are as follows

Abstract

- Remove “, compromising the reliability of results. ” from second line

- As authors are focusing on MLC, the first sentence should state why addressing missing values in MLC is important rather starting it from general perspective of missing value

- Define what is MLC in the abstract for the multidisciplinary readers of the journal

- This sentence is redundant to the focus of the manuscript “Multi-label learning is considered an emerging and promising research topic because of the growing number of new applications, such as the semantic classification of videos and images, music categorization, and medical diagnostics.” Remove it

- Use the full form for “EvoImp” before using it for example Evolutionary Imputation method (EvoImp)

- Add real results how EvoImp outperformed other methods and why it is innovative

- The method was compared with other state-of-the-art imputation strategies….. such as (add some names)

-

- Remove this “Following open-science principles, the source codes and datasets are publicly available in a GitHub repository.”

Introduction

- Remove these lines,

One of the most time-consuming tasks in the data mining pipeline is related to data 2 preparation. There is a consensus in this field that data preparation is responsible for 3 up to 80% of the entire process of discovering information [1]. In this case, selecting the 4 dataset is one of the first steps in the process and can help reduce the effort required in 5 the preprocessing phase [2].

- The author should start their introduction section with how missing values impact the scientific studies, decision making etc.

- Considering changing this sentence “Several techniques have emerged to address this problem. ”, as authors did not provided many examples here

- Line 15-17 Revise this sentence, the current sentence does not make any sense ….Another approach 15 that is quite widespread in the literature and non-trivial is the use of data imputation 16 methods that can be employed to search for what is regarded as plausible values to fill 17 any gaps that might be found [4,6–8].

- Line 20, Define what is Multiple Imputations (MI) before moving forward to more advanced imputation methods

- Line 30, “MultImp algorithm is coming out of nowhere, define it”

- Line 28-34: I suggest authors to define what is MLC, how GA-based imputation would have been used previously, what are the drawbacks of prior algorithms?

- Line 36-37: remove the sentence

- Line 47-49: Revise the sentence, the current format dose not make any sense. Also, the name refers to MultImp [12], on which the 47 algorithm was based, since it is a strategy that achieves promising results (although it is 48 still in the preliminary stage) concerning multiple imputations for missing data

- Author should add a paragraph talking about the drawbacks/limitation of the some existing MLC imputation methods, then start the methodology

Methods

- Line 66-68: Revise/rewrite it. “… Recently, some new applications are being investigated in the areas of Computer 66 Vision, Natural Language Processing and Data Mining (DM), such as Video 67 Annotation, Legal Text Mining and User Profiling [21]. “

- Line 104-105 Revise the sentences…. “All these challenges have increased the complexity when dealing with MVs. On the 104 other side, it is not easy to find studies that related MLC and MV, as seen in [2,11,29]. In this scenario, we highlight a few studies addressing the problem of missing 106 labels [30,31], i.e., focusing on predicting an unknown label. ”

- Line 110: change the word “relieve ”

- Line 111-112: merge two sentences: Cheng, Song & Qian [31] focus to dealing with missing labels using label correlations. 111 Therefore, the authors implements a two-level kernel extreme learning machine 112 autoencoder.

- Line 113: Use the authors instead of “They ”

- Line 115-174: I suggest creating a table to describe the bio-inspired algorithms for missing value imputations. The current paragraphs are lengthy and very difficult to follow. The table will include columns such as

Previous bioinspired algorithms| Methods | strengths| limitation

Add it as a supplementary material, so that the readers can go back to the literature review.

- Line 175: I suggest authors to develop a block diagram for each component of the GA structure:

Individual encoding and population initialization

Genetic Operators

Fitness Function

The Algorithm

Line 207- 212: Revise the sentences “Unlike MOGAImp [36], which uses random initialization of the initial population, the 207 proposed method relies on optimizing simple imputations through evolution to perform 208 multiple imputations. This strategy reduces the search space since it starts from a 209 priori solutions and represents the novelty compared to other methods. Furthermore, 210 this reduction allows for good use in scenarios where the computational cost for 211 calculating the objective function is sensitive, as in the case of multi-label classification. ”

Line 341-345: These sentences are very confusing, needs to be revised

“Regarding the simple imputation methods, the parameters recommended by [37] 341 were used. The mutation rate (MR) chosen is higher than the typical usage rates 342 because the starting point is not random. Therefore, considering that the initial 343 population is obtained by other methods, parameterization experiments demonstrated 344 that a higher MR yields better results, providing fast convergence. ”

Line 360-380: remove it to the supplementary file. Add one sentence in the main manuscript why we need to care about computational complexity for the proposed algorithm?

Table 5 and 6 7 8 9: Add footnotes by stating all the abbreviations of Db, KMI, KNNI MC< CMC, WKNNI ,EVoImp. What are these b, c, … ?? also explain what does it mean by uptick and downtick sign, what does the first column means- percentage of missingness? Then add it there

Discussion:

- Add a new paragraph summarizing strengths (at least 5) of the proposed imputation algorithm clearly compared to the prior algorithms

- Add a new paragraph summarizing limitations (at least 5) of the proposed imputation algorithm clearly

Reviewers' comments:

Reviewer's Responses to Questions

**Comments to the Author**

1. If the authors have adequately addressed your comments raised in a previous round of review and you feel that this manuscript is now acceptable for publication, you may indicate that here to bypass the “Comments to the Author” section, enter your conflict of interest statement in the “Confidential to Editor” section, and submit your "Accept" recommendation.

Reviewer #1: (No Response)

Reviewer #2: All comments have been addressed

2. Is the manuscript technically sound, and do the data support the conclusions?

Reviewer #1: Yes

Reviewer #2: Yes

3. Has the statistical analysis been performed appropriately and rigorously? 

Reviewer #1: Yes

Reviewer #2: Yes

4. Have the authors made all data underlying the findings in their manuscript fully available?

Reviewer #1: Yes

Reviewer #2: Yes

5. Is the manuscript presented in an intelligible fashion and written in standard English?

Reviewer #1: Yes

Reviewer #2: No

6. Review Comments to the Author

Reviewer #1: Author has revised my suggestion and I believe this article can be published。

Reviewer #2: The authors have provided a very thorough revision and have responded to all my comments. However, some grammatical errors or typos are shown indicatively below. Please check the whole manuscript again.

line 223: [41] provide → provides (or provided)

line 242: The table 2b → Table 2b

line 265: Exact Match: calculates → Exact Match calculates

line 330: 5, 10, 15, 20, 25, and 30% → 5%, 10%, …

In addition, a specific explanation of the mathematical notation needs to be given, such as in Eq. (3). Moreover, based on the description in the literature [54], it is recommended to redefine the mathematical expressions of Eqs. (5) and (6) to clarify the meaning of et al.

Overall, the revisions have strengthened the paper by bringing improved clarity, and focus on contributions.

7. PLOS authors have the option to publish the peer review history of their article (what does this mean?). If published, this will include your full peer review and any attached files.

Reviewer #1: No

Reviewer #2: No

---

## [Author Response · Author response to Decision Letter 1]

17 Nov 2023

Dear Editor,

We are grateful for the opportunity to revise and resubmit our paper. We have made sincere efforts to address all suggestions.

The file "PLOSONE - Response to Reviewers -v2023-11" presents detailed responses to the comments you and the two reviewers provided. The original comments are highlighted in bold, while our responses are presented in standard text.

We hope you find the quality of this paper has been improved.

Kind regards,

The Authors

---

## [Decision Letter · Decision Letter 2]

29 Dec 2023

EvoImp: Multiple Imputation of Multi-label classification data With a genetic algorithm

PONE-D-23-16570R2

Dear Dr. JACOB JUNIOR,

We’re pleased to inform you that your manuscript has been judged scientifically suitable for publication and will be formally accepted for publication once it meets all outstanding technical requirements.

Kind regards,

Mohammad A. Al-Mamun, PhD

Academic Editor

PLOS ONE

Additional Editor Comments (optional):

Reviewers' comments:

Reviewer's Responses to Questions

**Comments to the Author**

1. If the authors have adequately addressed your comments raised in a previous round of review and you feel that this manuscript is now acceptable for publication, you may indicate that here to bypass the “Comments to the Author” section, enter your conflict of interest statement in the “Confidential to Editor” section, and submit your "Accept" recommendation.

Reviewer #2: All comments have been addressed

2. Is the manuscript technically sound, and do the data support the conclusions?

Reviewer #2: Yes

3. Has the statistical analysis been performed appropriately and rigorously? 

Reviewer #2: Yes

4. Have the authors made all data underlying the findings in their manuscript fully available?

Reviewer #2: Yes

5. Is the manuscript presented in an intelligible fashion and written in standard English?

Reviewer #2: Yes

6. Review Comments to the Author

Reviewer #2: (No Response)

7. PLOS authors have the option to publish the peer review history of their article (what does this mean?). If published, this will include your full peer review and any attached files.

Reviewer #2: No

---

## [Editor Report · Acceptance letter]

9 Jan 2024

PONE-D-23-16570R2 

PLOS ONE

Dear Dr. Jacob Junior, 

I'm pleased to inform you that your manuscript has been deemed suitable for publication in PLOS ONE. Congratulations! Your manuscript is now being handed over to our production team.

Kind regards, 

on behalf of

Dr. Mohammad A. Al-Mamun 

Academic Editor

PLOS ONE